# Triangle and Four Cycle Counting with Predictions in Graph Streams

**Justin Y. Chen, Piotr Indyk, Shyam Narayanan, Ronitt Rubinfeld, and Sandeep Silwal** [*]
Computer Science and Artificial Intelligence Laboratory
Massachusetts Institute of Technology
Cambridge, MA 02139, USA
`{justc, indyk, shyamsn, ronitt, silwal}@mit.edu`

**Honghao Lin, David P. Woodruff, Michael Zhang**
Computer Science Department
Carnegie Mellon University
Pittsburgh, PA 15213, USA
`{honghaol, dwoodruf, jinyaoz}@andrew.cmu.edu`

**Tal Wagner**
Microsoft Research
Redmond, WA 98052, USA
`tal.wagner@gmail.com`

**Talya Eden**
Massachusetts Institute of Technology and Boston University
Cambridge, MA 02139, USA
`teden@mit.edu`

## ABSTRACT

We propose data-driven one-pass streaming algorithms for estimating the number of triangles and four cycles, two fundamental problems in graph analytics that are widely studied in the graph data stream literature. Recently, Hsu et al. (2019a) and Jiang et al. (2020) applied machine learning techniques in other data stream problems, using a trained oracle that can predict certain properties of the stream elements to improve on prior "classical" algorithms that did not use oracles. In this paper, we explore the power of a "heavy edge" oracle in multiple graph edge streaming models. In the adjacency list model, we present a one-pass triangle counting algorithm improving upon the previous space upper bounds without such an oracle. In the arbitrary order model, we present algorithms for both triangle and four cycle estimation with fewer passes and the same space complexity as in previous algorithms, and we show several of these bounds are optimal. We analyze our algorithms under several noise models, showing that the algorithms perform well even when the oracle errs. Our methodology expands upon prior work on "classical" streaming algorithms, as previous multi-pass and random order streaming algorithms can be seen as special cases of our algorithms, where the first pass or random order was used to implement the heavy edge oracle. Lastly, our experiments demonstrate advantages of the proposed method compared to state-of-the-art streaming algorithms.

## 1 INTRODUCTION

Counting the number of cycles in a graph is a fundamental problem in the graph stream model (e.g., Atserias et al. (2008); Bera & Chakrabarti (2017); Seshadhri et al. (2013); Kolountzakis et al. (2010); Bar-Yossef et al. (2002); Kallaugher et al. (2019)). The special case of counting triangles is widely studied, as it has a vast range of applications. In particular, it provides important insights into the structural properties of networks (Prat-Pérez et al., 2012; Farkas et al., 2011), and is used to discover motifs in protein interaction networks (Milo et al., 2002), understand social networks (Foucault Welles et al., 2010), and evaluate large graph models (Leskovec et al., 2008). See Al Hasan & Dave (2018) for a survey of these and other applications.

---

[*]All authors contributed equally.

Because of its importance, a large body of research has been devoted to space-efficient streaming algorithms for $(1 + \epsilon)$-approximate triangle counting. Such algorithms perform computation in one or few passes over the data using only a sub-linear amount of space. A common difficulty which arises in all previous works is the existence of *heavy edges*, i.e., edges that are incident to many triangles (four cycles). As sublinear space algorithms often rely on sampling of edges, and since a single heavy edge can greatly affect the number of triangles (four cycles) in a graph, sampling and storing these edges are often the key to an accurate estimation. Therefore, multiple techniques have been developed to determine whether a given edge is heavy or not.

Recently, based on the observation that many underlying patterns in real-world data sets do not change quickly over time, machine learning techniques have been incorporated into the data stream model via the training of heavy-hitter oracles. Given access to such a learning-based oracle, a wide range of significant problems in data stream processing — including frequency estimation, estimating the number of distinct elements, $F_p$-Moments or $(k, p)$-Cascaded Norms — can all achieve space bounds that are better than those provided by "classical" algorithms, see, e.g., Hsu et al. (2019a); Cohen et al. (2020); Jiang et al. (2020); Eden et al. (2021); Du et al. (2021). More generally, learning-based approaches have had wide success in other algorithmic tasks, such as data structures (Kraska et al., 2018; Ferragina et al., 2020; Mitzenmacher, 2018; Rae et al., 2019; Vaidya et al., 2021), online algorithms (Lykouris & Vassilvtiskii, 2018; Purohit et al., 2018; Gollapudi & Panigrahi, 2019; Rohatgi, 2020; Wei, 2020; Mitzenmacher, 2020; Lattanzi et al., 2020; Bamas et al., 2020), similarity search (Wang et al., 2016; Dong et al., 2020) and combinatorial optimization (Dai et al., 2017; Balcan et al., 2017; 2018a;b; 2019). See the survey and references therein for additional works (Mitzenmacher & Vassilvitskii, 2020).

Inspired by these recent advancements, we ask: *is it possible to utilize a learned heavy edge oracle to improve the space complexity of subgraph counting in the graph stream model?* Our results demonstrate that the answer is yes.

## 1.1 Our Results and Comparison to Previous Theoretical Works

We present theoretical and empirical results in several graph streaming models, and with several notions of prediction oracles. Conceptually, it is useful to begin by studying *perfect oracles* that provide exact predictions. While instructive theoretically, such oracles are typically not available in practice. We then extend our theoretical results to *noisy oracles* that can provide inaccurate or wrong predictions. We validate the practicality of such oracles in two ways: by directly showing they can be constructed for multiple real datasets, and by showing that on those datasets, our algorithms attain significant empirical improvements over baselines, when given access to these oracles.

We proceed to a precise account of our results. Let $G = (V, E)$ denote the input graph, and let $n$, $m$, and $T$ denote the number of vertices, edges, and triangles (or four-cycles) in $G$, respectively. There are two major graph edge streaming models: the *adjacency list* model and the *arbitrary order* model. We show that training heavy edge oracles is possible in practice in both models, and that such oracles make it possible to design new algorithms that significantly improve the space complexity of triangle and four-cycle counting, both in theory and in practice. Furthermore, our formalization of a heavy edge prediction framework makes it possible to show provable lower bounds as well. Our results are summarized in Table 1.

In our algorithms, we assume that we know a large-constant approximation of $T$ for the purposes of setting various parameters. This is standard practice in the subgraph counting streaming literature (e.g., see (Braverman et al., 2013, Section 1), (McGregor et al., 2016, Section 1.2) for an extensive discussions on this assumption). Moreover, when this assumption cannot be directly carried over in practice, in Subsection F.3 we discuss how to adapt our algorithms to overcome this issue.

### 1.1.1 Perfect Oracle

Our first results apply for the case that the algorithms are given access to a perfect heavy edge oracle. That is, for some threshold $\rho$, the oracle perfectly predicts whether or not a given edge is incident to at least $\rho$ triangles (four cycles). We describe how to relax this assumption in Section 1.1.2.

**Adjacency List Model** All edges incident to the same node arrive together. We show:

Table 1: Our results compared to existing theoretical algorithms. $\Delta_E$ ($\Delta_V$) denotes the maximum number of triangles incident to any edge (vertex), and $\kappa$ denotes the arboricity of the graph.

| Problem | Previous Results (no oracle) | | Our Results |
|---|---|---|---|
| Triangle, Adjacency | $\widetilde{O}(\epsilon^{-2}m/\sqrt{T})$, 1-pass (McGregor et al., 2016) | | $\widetilde{O}(\min(\epsilon^{-2}m^{2/3}/T^{1/3}, \epsilon^{-1}m^{1/2}))$, 1-pass |
| Triangle, Arbitrary | $\widetilde{O}(\epsilon^{-2}m^{3/2}/T)$, 3-pass (McGregor et al., 2016) | | $O(\epsilon^{-1}(m/\sqrt{T}+m^{1/2}))$, 1-pass |
| | $\widetilde{O}(\epsilon^{-2}m/\sqrt{T})$, 2-pass (McGregor et al., 2016) | | |
| | $\widetilde{O}(\epsilon^{-2}(m/\sqrt{T} + m\Delta_E/T))$, 1-pass (Pagh & Tsourakakis, 2012) | | |
| | $\widetilde{O}(\epsilon^{-2}(m/T^{2/3} + m\Delta_E/T + m\sqrt{\Delta_V}/T))$, 2-pass (Kallaugher & Price, 2017) | | |
| | $\widetilde{O}(\text{poly}(\epsilon^{-1})(m\Delta_E/T + m\sqrt{\Delta_V}/T))$, 1-pass (Jayaram & Kallaugher, 2021) | | |
| | $\widetilde{O}(\text{poly}(\epsilon^{-1})m\kappa/T)$, multi-pass (Bera & Sheshadhri, 2020) | | |
| 4-cycle, Arbitrary | $\widetilde{O}(\epsilon^{-2}m/T^{1/4})$, 3-pass (McGregor & Vorotnikova, 2020) | | $\widetilde{O}(T^{1/3} + \epsilon^{-2}m/T^{1/3})$, 1-pass |
| | $\widetilde{O}(\epsilon^{-2}m/T^{1/3})$, 3-pass (Vorotnikova, 2020) | | |

**Theorem 1.1.** *There exists a one-pass algorithm, Algorithm 1, with space complexity*[1] $\widetilde{O}(\min(\epsilon^{-2}m^{2/3}/T^{1/3}, \epsilon^{-1}m^{1/2}))$ *in the adjacency list model that, using a learning-based oracle, returns a* $(1 \pm \epsilon)$-*approximation to the number $T$ of triangles with probability at least*[2] $7/10$.

An overview of Algorithm 1 is given in Section 2, and the full analysis is provided in Appendix B.

**Arbitrary Order Model** In this model, the edges arrive in the stream in an arbitrary order. We present a one-pass algorithm for triangle counting and another one-pass algorithm for four cycle counting in this model, both reducing the number of passes compared to the currently best known space complexity algorithms. Our next result is as follows:

**Theorem 1.2.** *There exists a one-pass algorithm, Algorithm 4, with space complexity* $\widetilde{O}(\epsilon^{-1}(m/\sqrt{T} + \sqrt{m}))$ *in the arbitrary order model that, using a learning-based oracle, returns a* $(1 \pm \epsilon)$-*approximation to the number $T$ of triangles with probability at least* $7/10$.

An overview of Algorithm 4 is given in Section 3, and full details are provided in Appendix C.

We also show non-trivial space lower bounds that hold even if appropriate predictors are available. In Theorem C.2 in Appendix C.3, we provide a lower bound for this setting by giving a construction that requires $\Omega(\min(m/\sqrt{T}, m^{3/2}/T))$ space even with the help of an oracle, proving that our result is nearly tight in some regimes. Therefore, the triangle counting problem remains non-trivial even when extra information is available.

**Four Cycle Counting.** For four cycle counting in the arbitrary order model, we give Theorem 1.3 which is proven in Appendix D.

**Theorem 1.3.** *There exists a one-pass algorithm, Algorithm 5, with space complexity* $\widetilde{O}(T^{1/3} + \epsilon^{-2}m/T^{1/3})$ *in the arbitrary order model that, using a learning-based oracle, returns a* $(1 \pm \epsilon)$-*approximation to the number $T$ of four cycles with probability at least* $7/10$.

To summarize our theoretical contributions, for the first set of results of counting triangles in the adjacency list model, our bounds always improve on the previous state of the art due to McGregor et al. (2016) for all values of $m$ and $T$. For a concrete example, consider the case that $T = \Theta(\sqrt{m})$.

---

[1]We use $\widetilde{O}(f)$ to denote $O(f \cdot \text{polylog}(f))$.
[2]The success probability can be $1 - \delta$ by running $\log(1/\delta)$ copies of the algorithm and taking the median.

In this setting previous bounds result in an $\widetilde{O}(m^{3/4})$-space algorithm, while our algorithm only requires $\widetilde{O}(\sqrt{m})$ space (for constant $\epsilon$).

For the other two problems of counting triangles and 4-cycles in the arbitrary arrival model, our space bounds have an additional additive term compared to McGregor et al. (2016) (for triangles) and Vorotnikova (2020) (for 4-cycles) but importantly run in a **single pass** rather than multiple passes. In the case where the input graph has high triangles density, $T = \Omega(m/\epsilon^2)$, our space bound is worse due to the additive factor. When $T = O(m/\epsilon^2)$, our results achieve the same dependence on $m$ and $T$ as that of the previous algorithms with an improved dependency in $\epsilon$. Moreover, the case $T \leq m/\epsilon^2$ is natural for many real world datasets: for $\epsilon = 0.05$, this condition holds for all of the datasets in our experimental results (see Table 2). Regardless of the triangle density, a key benefit of our results is that they are achieved in a single pass rather than multiple passes. Finally, our results are for general graphs, and make no assumptions on the input graph (unlike Pagh & Tsourakakis (2012); Kallaugher & Price (2017); Bera & Sheshadhri (2020)). Most of our algorithms are relatively simple and easy to implement and deploy. At the same time, some of our results require the use of novel techniques in this context, such as the use of exponential random variables (see Section 2).

### 1.1.2 Noisy Oracles

The aforementioned triangle counting results are stated under the assumption that the algorithms are given access to perfect heavy edge oracles. In practice, this assumption is sometimes unrealistic. Hence, we consider several types of noisy oracles. The first such oracle, which we refer to as a $K$-noisy oracle, is defined below (see Figure 3 in the Supplementary Section C.2).

**Definition 1.1.** *For an edge $e = xy$ in the stream, define $N_e$ as the number of triangles that contain both $x$ and $y$. For a fixed constant $K \geq 1$ and for a threshold $\rho$ we say that an oracle $O_\rho$ is a $K$-noisy oracle if for every edge $e$, $1 - K \cdot \frac{\rho}{N_e} \leq Pr[O_\rho(e) = \text{HEAVY}] \leq K \cdot \frac{N_e}{\rho}$.*

This oracle ensures that if an edge is extremely heavy or extremely light, it is classified correctly with high probability, but if the edge is close to the threshold, the oracle may be inaccurate. We further discuss the properties of this oracle in Section G.

For this oracle, we prove the following two theorems. First, in the adjacency list model, we prove:

**Theorem 1.4.** *Suppose that the oracle given to Algorithm 1 is a $K$-noisy oracle as defined in Definition 1.1. Then with probability 2/3, Algorithm 1 returns a value in $(1 \pm \sqrt{K} \cdot \epsilon)T$, and uses space at most $\widetilde{O}(\min(\epsilon^{-2}m^{2/3}/T^{1/3}, K \cdot \epsilon^{-1}m^{1/2}))$.*

Hence, even if our oracle is inaccurate for edges near the threshold, our algorithm still obtains an effective approximation with low space in the adjacency list model. Likewise, for the arbitrary order model, we prove in Theorem C.1 that the $O(\epsilon^{-1}(m/\sqrt{T} + \sqrt{m}))$ 1-pass algorithm of Theorem 1.2 also works when Algorithm 4 is only given access to a $K$-noisy oracle.

The proof of Theorem 1.4 is provided in Appendix B.1, and the proof of Theorem C.1 is provided in Appendix C.2. We remark that Theorems 1.4 and C.1 automatically imply Theorems 1.1 and 1.2, since the perfect oracle is automatically a $K$-noisy oracle.

**(Noisy) Value Oracles** In the adjacency list model, when we see an edge $xy$, we also have access to all the neighbors of either $x$ or $y$, which makes it possible for the oracle to give a more accurate prediction. For an edge $xy$, let $R_{xy}$ denote the number of triangles $\{x, z, y\}$ so that $x$ precedes $z$ and $z$ precedes $y$ in the stream arrival order. Formally, $R_{xy} = |z : \{x, y, z\} \in \Delta$ and $x <_s z <_s y|$ where $x <_s y$ denotes that the adjacency list of $x$ arrives before that of $y$ in the stream.

Motivated by our empirical results in Section F.7, it is reasonable in some settings to assume we have access to oracles that can predict a good approximation to $R_{xy}$. We refer to such oracles as *value oracles*.

In the first version of this oracle, we assume that the probability of approximation error decays linearly with the error from above but exponentially with the error from below.

**Definition 1.2.** *Given an edge $e$, an $(\alpha, \beta)$ value-prediction oracle outputs a random value $p(e)$ where $\mathbb{E}[p(e)] \leq \alpha R_e + \beta$, and $\mathbf{Pr}[p(e) < \frac{R_e}{\lambda} - \beta] \leq Ke^{-\lambda}$ for some constant $K$ and any $\lambda \geq 1$.*

For this variant, we prove the following theorem.

**Theorem 1.5.** *Given an oracle with parameters $(\alpha, \beta)$, there exists a one-pass algorithm, Algorithm 2, with space complexity $O(\epsilon^{-2} \log^2(K/\epsilon)(\alpha + m\beta/T))$ in the adjacency list model that returns a $(1 \pm \epsilon)$-approximation to the number of triangles $T$ with probability at least $7/10$.*

In the second version of this noisy oracle, we assume that the probability of approximation error decays linearly with the error from both above and below. For this variant, we prove that we can achieve the same guarantees as Theorem 1.5 up to logarithmic factors (see Theorem B.1). The algorithms and proofs for both Theorem 1.5 and Theorem B.1 appear in Appendix B.2.

**Experiments** We conduct experiments to verify our results for triangle counting on a variety of real world networks (see Table 2) in both the arbitrary and adjacency list models. Our algorithms use additional information through predictors to improve empirical performance. The predictors are data dependent and include: memorizing heavy edges in a small portion of the first graph in a sequence of graphs, linear regression, and graph neural networks (GNNs). Our experimental results show that we can achieve up to **5x** decrease in estimation error while keeping the same amount of edges as other state of the art empirical algorithms. For more details, see Section 4. In Section F.7, we show that our noisy oracle models are realistic for real datasets.

**Related Empirical Works** On the empirical side, most of the focus has been on triangle counting in the arbitrary order model for which there are several algorithms that work well in practice. We primarily focus on two state-of-the-art baselines, ThinkD (Shin et al., 2018) and WRS (Shin, 2017). In these works, the authors compare to previous empirical benchmarks such as the ones given in Stefani et al. (2017); Han & Sethu (2017); Lim & Kang (2015) and demonstrate that their algorithms achieve superior estimates over these benchmarks. There are also other empirical works such as Ahmed et al. (2017) and Ahmed & Duffield (2020) studying this model but they do not compare to either ThinkD or WRS. While these empirical papers demonstrate that their algorithm returns unbiased estimates, their theoretical guarantees on space is incomparable to the previously stated space bounds for theoretical algorithms in Table 1. Nevertheless, we use ThinkD and WRS as part of our benchmarks due to their strong practical performance and code accessibility.

**Implicit Predictors in Prior Works** The idea of using a predictor is implicit in many prior works. The optimal two pass triangle counting algorithm of McGregor et al. (2016) can be viewed as an implementation of a heavy edge oracle after the first pass. This oracle is even stronger than the $K$-noisy oracle as it is equivalent to an oracle that is always correct on an edge $e$ if $N_e$ either exceeds or is under the threshold $\rho$ by a constant multiplicative factor. This further supports our choice of oracles in our theoretical results, as a stronger version of our oracle can be implemented using one additional pass through the data stream (see Section G). Similarly, the optimal triangle counting streaming algorithm (assuming a *random* order) given in McGregor & Vorotnikova (2020) also implicitly defines a heavy edge oracle using a small initial portion of the random stream (see Lemma 2.2 in McGregor & Vorotnikova (2020)). The random order assumption allows for the creation of such an oracle since heavy edges are likely to have many of their incident triangle edges appearing in an initial portion of the stream. We view these two prior works as theoretical justification for our oracle definitions. Lastly, the WRS algorithm also shares the feature of defining an implicit oracle: some space is reserved for keeping the most recent edges while the rest is used to keep a random sample of edges. This can be viewed as a specific variant of our model, where the oracle predicts recent edges as heavy.

**Preliminaries.** $G = (V, E)$ denotes the input graph, and $n$, $m$ and $T$ denote the number of vertices, edges and triangles (or four-cycles) in $G$, respectively. We use $N(v)$ to denote the set of neighbors of a node $v$, and $\Delta$ to denote the set of triangles. In triangle counting, for each $xy \in E(G)$, we recall that $N_{xy} = |z : \{x, y, z\} \in \Delta|$ is the number of triangles incident to edge $xy$, and $R_{xy} = |z : \{x, y, z\} \in \Delta, x <_s z <_s y|$ is the number of triangles adjacent to $xy$ with the third vertex $z$ of the triangle between $x$ and $y$ in the adjacency list order. Table A summarizes the notation.

## 2 TRIANGLE COUNTING IN THE ADJACENCY LIST MODEL

We describe an algorithm with a heavy edge oracle, and one with a value oracle.

**Heavy Edge Oracle.** We present an overview of our one-pass algorithm, Algorithm 1, with a space complexity of $\widetilde{O}(\min(\epsilon^{-2} m^{2/3}/T^{1/3}, \epsilon^{-1} m^{1/2}))$, given in Theorem 1.1. We defer the pseudocode

and proof of the theorem to Appendix B. The adaptations and proofs for Theorems 1.4, 1.5 and B.1 for the various noisy oracles appear in Appendix B.1 and Appendix B.2.

Our algorithm works differently depending on the value of $T$. We first consider the case that $T \geq (m/\epsilon)^{1/2}$. Assume that for each sampled edge $xy$ in the stream, we can exactly know the number of triangles $R_{xy}$ this edge contributes to $T$. Then the rate at which we would need to sample each edge would be proportional to $p_{naive} \approx \epsilon^{-2} \Delta_E / T$, where $\Delta_E$ is the maximum number of triangles incident to any edge. Hence, our first idea is to separately consider light and non-light edges using the heavy edge oracle. This allows us to sample edges that are deemed light by the oracle at a lower rate, $p_1$, and compute their contribution by keeping track of $R_{xy}$ for each such sampled edge. Intuitively, light edges offer us more flexibility and thus we can sample them with a lower probability while ensuring the estimator's error does not drastically increase. In order to estimate the contribution due to non-light edges, we again partition them into two types: medium and heavy, according to some threshold $\rho$. We then use an observation from McGregor et al. (2016), that since for heavy edges $R_{xy} > \rho$, it is sufficient to sample from the entire stream at rate $p_3 \approx \epsilon^{-2}/\rho$, in order to both detect if some edge $xy$ is heavy and if so to estimate $R_{xy}$.

Therefore, it remains to estimate the contribution to $T$ due to medium edges (these are the edges that are deemed non-light by the oracle, and also non-heavy according to the sub-sampling above). Since the number of triangles incident to medium edges is higher than that of light ones, we have to sample them at some higher rate $p_2 > p_1$. However, since their number is bounded, this is still space efficient. We get the desired bounds by correctly setting the thresholds between light, medium and heavy edges.

When $T < (m/\epsilon)^{1/2}$ our algorithm becomes much simpler. We only consider two types of edges, light and heavy, according to some threshold $T/\rho$. To estimate the contribution due to heavy edges we simply store them and keep track of their $R_{xy}$ values. To estimate the contribution due to light edges we sub-sample them with rate $\epsilon^{-2} \cdot \Delta_E / T = \epsilon^{-2}/\rho$. The total space we use is $\widetilde{O}(\epsilon^{-1}\sqrt{m})$, which is optimal in this case. See Algorithm 1 for more details of the implementation.

**Value-Based Oracle.** We also consider the setting where the predictor returns an estimate $p(e)$ of $R_e$, and we assume $R_e \leq p(e) \leq \alpha \cdot R_e$, where $\alpha \geq 1$ is an approximation factor. We relax this assumption to also handle additive error as well as noise, but for intuition we focus on this case. The value-based oracle setting requires the use of novel techniques, such as the use of exponential random variables (ERVs). Given this oracle, for an edge $e$, we compute $p(e)/u_e$, were $u_e$ is a standard ERV. We then store the $O(\alpha \log(1/\epsilon))$ edges $e$ for which $p(e)/u_e$ is largest. Since we are in the adjacency list model, once we start tracking edge $e = xy$, we can also compute the true value $R_e$ of triangles that the edge $e$ participates in (since for each future vertex $z$ we see, we can check if $x$ and $y$ are both neighbors of $z$). Note that we track this quantity only for the $O(\alpha \log(1/\epsilon))$ edges that we store. Using the max-stability property of ERVs, $\max_e R_e/u_e$ is equal in distribution to $T/u$, where $u$ is another ERV. Importantly, using the density function of an ERV, one can show that the edge $e$ for which $R_e/u_e$ is largest is, with probability $1 - O(\epsilon^3)$, in our list of the $O(\alpha \log(1/\epsilon))$ largest $p(e)/u_e$ values that we store. Repeating this scheme $r = O(1/\epsilon^2)$ times, we obtain independent estimates $T/u^1, \ldots, T/u^r$, where $u^1, \ldots, u^r$ are independent ERVs. Taking the median of these then gives a $(1 \pm \epsilon)$-approximation to the total number $T$ of triangles. We note that ERVs are often used in data stream applications (see, e.g., Andoni (2017)), though to the best of our knowledge they have not previously been used in the context of triangle estimation. We also give an alternative algorithm, based on subsampling at $O(\log n)$ scales, which has worse logarithmic factors in theory but performs well empirically.

## 3 TRIANGLE COUNTING IN THE ARBITRARY ORDER MODEL

In this section we discuss Algorithm 4 for estimating the number of triangles in an arbitrary order stream of edges. The pseudo-code of the algorithm as well as omitted proofs for the different oracles are given in Supplementary Section C. Here we give the intuition behind the algorithm. Our approach relies on sampling the edges of the stream as they arrive and checking if every new edge forms a triangle with the previously sampled edges. However, as previously discussed, this approach alone fails if some edges have a large number of triangles incident to them as "overlooking" such edges might lead to an underestimation of the number of triangles. Therefore, we utilize a heavy edge oracle, and refer to edges that are not heavy as *light*. Whenever a new edge arrives, we first

query the oracle to determine if the edge is heavy. If the edge is heavy we keep it, and otherwise we sample it with some probability. As in the adjacency list arrival model case, this strategy allows us to reduce the variance of our estimator. By balancing the sampling rate and our threshold for heaviness, we ensure that the space requirement is not too high, while simultaneously guaranteeing that our estimate is accurate.

In more detail, our algorithm works as follows. First, we set a heaviness threshold $\rho$, so that if we predict an edge $e$ to be part of $\rho$ or more triangles, we label it as heavy. We also set a sampling parameter $p$. We let $H$ be the set of edges predicted to be heavy and let $S_L$ be a random sample of edges predicted to be light. Then, we count three types of triangles. The first counter, $\ell_1$, counts the triangles $\Delta = (e_1, e_2, e)$, where the first two edges seen in this triangle by the algorithm, represented by $e_1$ and $e_2$, are both in $S_L$. Note that we only count the triangle if $e_1$ and $e_2$, were both in $S_L$ at the time $e$ arrives in the stream. Similarly, $\ell_2$ counts triangles whose first two edges are in $S_L$ and $H$ (in either order), and $\ell_3$ counts triangles whose first two edges are in $H$. Finally, we return the estimate $\ell = \ell_1/p^2 + \ell_2/p + \ell_3$. Note that if the first two edges in any triangle are light, they will both be in $S_L$ with probability $p^2$, and if exactly one of the first two edges is light, it will be in $S_L$ with probability $p$. Therefore, we divide $\ell_1$ by $p^2$ and $\ell_2$ by $p$ so that $\ell$ is an unbiased estimator.

## 4 EXPERIMENTS

We now evaluate our algorithm on real and synthetic data whose properties are summarized in Table 2 (see Appendix F for more details).

Table 2: Datasets used in our experiments. Snapshot graphs are a sequence of graphs over time (the length of the sequence is given in parentheses) and temporal graphs are formed by edges appearing over time. The listed values for $n$ (number of vertices), $m$ (number of edges), and $T$ (number of triangles) for Oregon and CAIDA are approximated across all graphs. The Oregon and CAIDA datasets come from Leskovec & Krevl (2014); Leskovec et al. (2005), the Wikibooks dataset comes from Rossi & Ahmed (2015), the Reddit dataset comes from Leskovec & Krevl (2014); Kumar et al. (2018), the Twitch dataset comes from Rozemberczki et al. (2019), the Wikipedia dataset comes from Rossi & Ahmed (2015), and the Powerlaw graphs are sampled from the Chung-Lu-Vu random graph model with expected degree of the $i$-th vertex proportional to $1/i^2$ (Chung et al., 2003).

| Name | Type | Predictor | $n$ | $m$ | $T$ |
|------|------|-----------|-----|-----|-----|
| Oregon | Snapshot (9) | 1st graph | $\sim 10^4$ | $\sim 2.2 \cdot 10^4$ | $\sim 1.8 \cdot 10^4$ |
| CAIDA 2006 | Snapshot (52) | 1st graph | $\sim 2.2 \cdot 10^4$ | $\sim 4.5 \cdot 10^4$ | $\sim 3.4 \cdot 10^4$ |
| CAIDA 2007 | Snapshot (46) | 1st graph | $\sim 2.5 \cdot 10^4$ | $\sim 5.1 \cdot 10^4$ | $\sim 3.9 \cdot 10^4$ |
| Wikibooks | Temporal | Prefix | $\sim 1.3 \cdot 10^5$ | $\sim 3.9 \cdot 10^5$ | $\sim 1.8 \cdot 10^5$ |
| Reddit | Temporal | Regression | $\sim 3.6 \cdot 10^4$ | $\sim 1.2 \cdot 10^5$ | $\sim 4.1 \cdot 10^5$ |
| Twitch | - | GNN | $\sim 6.5 \cdot 10^3$ | $\sim 5.7 \cdot 10^4$ | $\sim 5.4 \cdot 10^4$ |
| Wikipedia | - | GNN | $\sim 4.8 \cdot 10^3$ | $\sim 4.6 \cdot 10^4$ | $\sim 9.5 \cdot 10^4$ |
| Powerlaw | Synthetic | EV | $\sim 1.7 \cdot 10^5$ | $\sim 10^6$ | $\sim 3.9 \cdot 10^7$ |

We now describe the edge heaviness predictors that we use (see also Table 2). Our predictors adapt to the type of dataset and information available for each dataset. Some datasets we use contain only the graph structure (nodes and edges) without semantic features, thus not enabling us to train a classical machine learning predictor for edge heaviness. In those cases we use the true counts on a small prefix of the data (either $10\%$ of the first graph in a sequence of graphs, or a prefix of edges in a temporal graph) as predicted counts for subsequent data. However, we are able to create more sophisticated predictors on three of our datasets, using feature vectors in linear regression or a Graph Neural Network. Precise details of the predictors follow.

· **Snapshot**: For Oregon / CAIDA graph datasets, which contain a sequence of graphs, we use exact counting on a *small* fraction of the first graph as the predictor for *all the subsequent graphs*. Specifically, we count the number of triangles per edge, $N_e$, on the first graph for each snapshot dataset. We then only store $10\%$ of the top heaviest edges and use these values as estimates for edge heaviness in all later graphs. If a queried edge is not stored, its predicted $N_e$ value is 0.

- **Prefix**: In the WikiBooks temporal graph, we use the exact $N_e$ counts on the first half of the graph edges (when sorted by their timestamps) as the predicted values for the second half.

- **Linear Regression**: In the Reddit Hyperlinks temporal graph, we use a separate dataset (Kumar et al., 2019) that contains 300-dimensional feature embeddings of subreddits (graph nodes). Two embeddings are close in the feature space if their associated subreddits have similar sub-communities. To produce an edge $f(e)$ embedding for an edge $e = uv$ from the node embedding of its endpoints, $f(u)$ and $f(v)$, we use the 602-dimensional embedding $(f(u), f(v), \|(f(u) - f(v)\|_1, \|(f(u) - f(v)\|_2)$. We then train a linear regressor to predict $N_e$ given the edge embedding $f(e)$. Training is done on a prefix of the first half of the edges.

- **Link Prediction (GNN)**: For each of the two networks, we start with a graph that has twice as many edges as listed in Table 2, ($\sim 1.1 \cdot 10^5$ edges for Twitch and $9.2 \cdot 10^4$ edges for Wikipedia). We then randomly remove $50\%$ of the total edges to be the training data set, and use the remaining edges as the graph we test on. We use the method proposed in Zhang & Chen (2018) to train a link prediction oracle using a Graph Neural Network (GNN) that will be used to predict the heaviness of the testing edges. For each edge that arrives in the stream of the test edges, we use the predicted likelihood of forming an edge given by the the neural network to the other vertices as our estimate for $N_{uv}$, the number of triangles on edge $uv$. See Section F.2 for details of training methodology.

- **Expected Value (EV)**: In the Powerlaw graph, the predicted number of triangles incident to each $N_e$ is its expected value, which can be computed analytically in the CLV random graph model.

**Baselines.** We compare our algorithms with the following baselines.

- **ThinkD and WRS** (Arbitrary Order): These are the state of the art empirical one-pass algorithms from Shin et al. (2018) and Shin (2017) respectively. The ThinkD paper presents two versions of the algorithm, called 'fast' and 'accurate'. We use the 'accurate' version since it provides better estimates. We use the authors' code for our experiments (Shin et al., 2020; Shin, 2020).

- **MVV** (Arbitrary Order and Adjacency List): We use the one pass streaming algorithms given in McGregor et al. (2016) for the arbitrary order model and the adjacency list model.

**Error measurement.** We measure accuracy using the relative error $|1 - \widetilde{T}/T|$, where $T$ is the true triangle count and $\widetilde{T}$ is the estimate returned by an algorithm. Our plots show the space used by an algorithm (in terms of the number of edges) versus the relative error. We report median errors over $50$ independent executions of each experiment, $\pm$ one standard deviation.

## 4.1 RESULTS FOR ARBITRARY ORDER TRIANGLE COUNTING EXPERIMENTS

In this section, we give experimental results for Algorithm 4 which approximates the triangle count in arbitrary order streams. Note that we need to set two parameters for Algorithm 4: $p$, which is the edge sampling probability, and $\rho$, which is the heaviness threshold. In our theoretical analysis, we assume knowledge of a lower bound on $T$ in order to set $p$ and $\rho$, as is standard in the theoretical streaming literature. However, in practice, such an estimate may not be available; in most cases, the only parameter we are given is a space bound for the number of edges that can be stored. To remedy this discrepancy, we modify our algorithm slightly by setting a fixed fraction of space to use for heavy edges ($10\%$ of space for all of our experiments) and setting $p$ correspondingly to use up the rest of the space bound given as input. See details in Supplementary Section F.3.

**Oregon and CAIDA** In Figures 1(a) and 1(b), we display the relative error as a function of increasing space for graph #4 in the dataset for Oregon and graph #30 for CAIDA 2006. These figures show that our algorithm outperforms the other baselines by as much as a **factor of 5**. We do not display the error bars for MVV and WRS for the sake of visual clarity, but they are comparable to or larger than both ThinkD and our algorithm. A similar remark applies to all figures in Figure 1. As shown in Figure 2, these specific examples are reflective of the performance of our algorithm across the whole sequence of graphs for Oregon and CAIDA. We also show qualitatively similar results for CAIDA 2007 in Figure 4(a) in Supplementary Section F.4.

We also present accuracy results over the various graphs of the Oregon and CAIDA datasets. We fix the space to be $10\%$ of the number of edges (which varies across graphs). Our results for the Oregon dataset are plotted in Figure 2(a) and the results for CAIDA 2006 and 2007 are plotted in

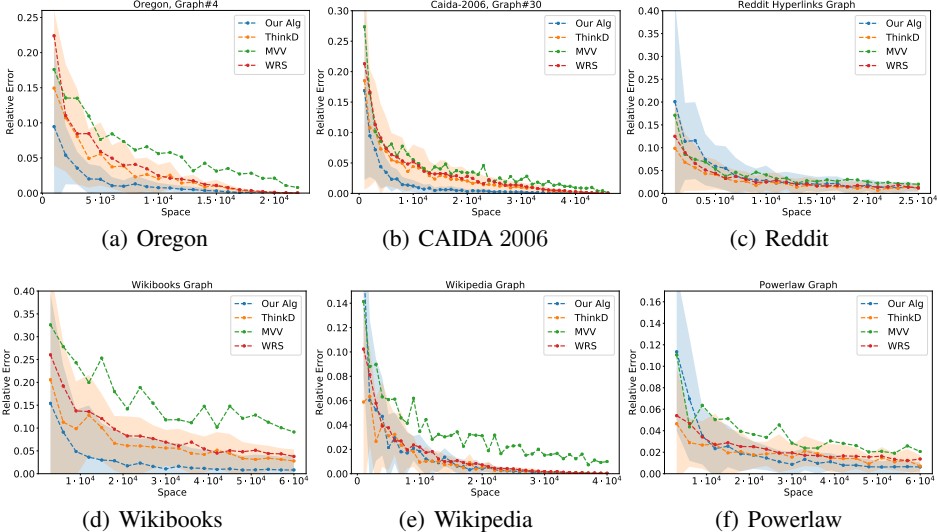

Figure 1: Error as a function of space in the arbitrary order model.

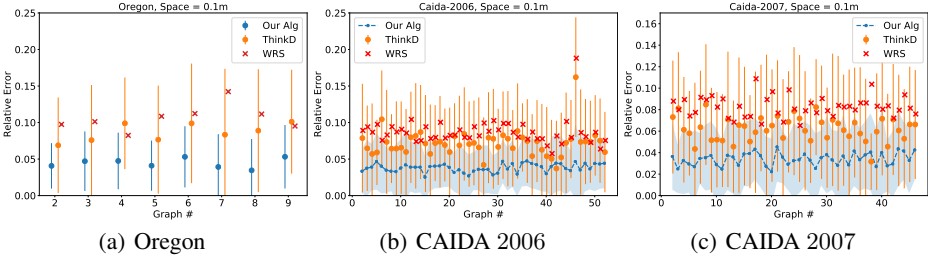

Figure 2: Error across snapshot graphs with space $0.1m$ in the arbitrary order model.

Figures 2(b) and 2(c), respectively. These figures illustrate that the quality of the predictor remains consistent over time even after a year has elapsed between when the first and the last graphs were created as our algorithm outperforms the baselines on average by up to a factor of 2.

**Reddit** Our results are displayed in Figure 1(c). All four algorithms are comparable as we vary space. While we do not improve over baselines in this case, this dataset serves to highlight the fact that predictors can be trained using node or edge semantic information (i.e., features).

**Wikibooks and Powerlaw** For Wikibooks, we see in Figure 1(d) that our algorithm is outperforming ThinkD and WRS by at least a factor of 2, and these algorithms heavily outperform the MVV algorithm. Nonetheless, it is important to note that our algorithm uses the exact counts on the first half of the edges (encoded in the predictor), which encode a lot information not available to the baselines. Thus the takeaway is that in temporal graphs, where edges arrive continuously over time (e.g., citation networks, road networks, etc.), using a prefix of the edges to form noisy predictors can lead to a significant advantage in handling future data on the same graph. Our results for Powerlaw are presented in Figure 1(f). Here too we see that as the space allocation increases, our algorithm outperforms ThinkD, MVV, and WRS.

**Twitch and Wikipedia** In Figure 1(e), we see that three algorithms, ours, MVV, and WRS, are all comparable as we vary space. The qualitatively similar result for Twitch is given in Figure 4(b). These datasets serve to highlight that predictors can be trained using modern ML techniques such as Graph Neural Networks. Nevertheless, these predictors help our algorithms improve over baselines for experiments in the adjacency list model (see Section below).

**Results for Adjacency List Experiments** Our experimental results for adjacency list experiments, which are qualitatively similar to the arbitrary order experiments, are given in full detail in Sections F.5 and F.6.

ACKNOWLEDGMENTS

Justin is supported by the NSF Graduate Research Fellowship under Grant No. 1745302 and Math-Works Engineering Fellowship. Sandeep and Shyam are supported by the NSF Graduate Research Fellowship under Grant No. 1745302. Ronitt was supported by NSF awards CCF-2006664, DMS 2022448, and CCF-1740751. Piotr was supported by the NSF TRIPODS program (awards CCF-1740751 and DMS-2022448), NSF award CCF-2006798 and Simons Investigator Award. Talya is supported in part by the NSF TRIPODS program, award CCF-1740751 and Ben Gurion University Postdoctoral Scholarship. Honghao Lin and David Woodruff would like to thank for partial support from the National Science Foundation (NSF) under Grant No. CCF-1815840.

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

# A  NOTATIONS TABLE

| Notation | Definition |
|---|---|
| $G = (V, E)$ | a graph $G$ with vertex set $V$ and edge set $E$ |
| $n$ | number of triangles |
| $m$ | number of edges |
| $T$ | number of triangles (or 4-cycles) |
| $\epsilon$ | approximation parameter |
| $<_s$ | total ordering on the vertices according to their arrival in the adjacency list arrival model |
| $\Delta$ | set of triangles in $G$ |
| $\square$ | set of 4-cycles in $G$ |
| $N(v)$ | set of neighbors of node $v$ |
| $N_{xy}$ | number of triangles (4-cycles) incident to edge $xy$, i.e., $\|\{z\|(x,y,z) \in \Delta\}\|$  ( $\|\{w,z\|(x,y,w.z) \in \square\}\|$ ) |
| $R_{xy}$ | number of triangles $(x,z,y)$ where $x$ precedes $z$ and $z$ precedes $y$ in the adjacency list arrival model, i.e., $\|\{z\|(x,z,y) \in \Delta, x <_s z <_s y\}\|$ |
| $\Delta_E$ | maximum number of triangles incident to any edge |
| $\Delta_V$ | maximum number of triangles incident to any vertex |
| $O_\rho$ | heavy edge oracle with threshold $\rho$ |

# B  OMITTED PROOFS FROM SECTION 2

In this section, we prove correctness and bound the space of Algorithm 1, thus proving Theorem 1.1. In the adjacency list model, recall that we have a total ordering on nodes $<_s$ based on the stream ordering, where $x <_s y$ if and only if the adjacency list of $x$ arrives before the adjacency list of $y$ in the stream. For each $xy \in E(G)$, we define $R_{xy} = |z : \{x,y,z\} \in \Delta$ and $x <_s z <_s y|$. Note that if $y <_s x$, then $R_{xy} = 0$. It holds that $\sum_{xy \in E} R_{xy} = T$.

We first give a detailed overview of the algorithm. We first consider the case when $T \geq (m/\epsilon)^{1/2}$. Let $\rho = (mT)^{1/3}$. We define the edge $xy$ to be *heavy* if $R_{xy} \geq \rho$, *light* if $R_{xy} \leq \frac{T}{\rho}$, and *medium* if $\frac{T}{\rho} < R_{xy} < \rho$. We train the oracle to predict, upon seeing $xy$, whether $R_{xy}$ is light or not. We define $H, M$, and $L$ as the set of heavy, medium, and light edges, respectively. Define $T_L = \sum_{xy \in L} R_{xy}$. We define $T_H$ and $T_M$ similarly. Since $\sum_{xy \in E} R_{xy} = T = T_L + T_M + T_H$, it suffices to estimate each of these three terms.

For the light edges, as shown in Lemma B.2, if we sample edges with rate $p_1 \approx \epsilon^{-2}/\rho$, with high probability we obtain an estimate of $T_L$ within error $\pm \epsilon T$. For the heavy edges, we recall an observation in McGregor et al. (2016): for an edge $xy$ where $R_{xy}$ is large, even if we do not sample $xy$ directly, if we sample from the stream (at rate $p_3 \approx \epsilon^{-2}/\rho$), we are likely to sample some edges in the set $|xz : \{x,y,z\} \in \Delta$ and $x <_s z <_s y|$. We refer to the sampled set of these edges as $S_{aux}$.

Further, we will show that not only can we use $S_{aux}$ to recognize whether $R_{xy}$ is large, but also to estimate $R_{xy}$ (Lemma B.3). What remains to handle is the medium edges. Since medium edges have higher values of $R_{xy}$ than light edges, we sample with a larger rate, $p_2$, to reduce variance. However, we show there cannot be too many medium edges, so we do not pay too much extra storage space. The overall space we use is $\widetilde{O}(\epsilon^{-2} m^{2/3}/T^{1/3})$.

When $T < (m/\epsilon)^{1/2}$ our algorithm becomes much simpler. Let $\rho = \sqrt{m}/\epsilon$. We define the edge $xy$ to be *heavy* if $R_{xy} \geq T/\rho$, and *light* otherwise (so in this case all the medium edges become heavy

edges). We directly store all the heavy edges and sub-sample with rate $\epsilon^{-2}/\rho$ to estimate $T_L$. The total space we use is $\widetilde{O}(\epsilon^{-1}\sqrt{m})$, which is optimal in this case. See Algorithm 1 for more details of the implementation.

---

**Algorithm 1** Counting Triangles in the Adjacency List Model

---

1: **Input:** Adjacency list edge stream and an oracle $O$ that outputs HEAVY if $R_{xy} \geq T/\rho$ and LIGHT otherwise.
2: **Output:** An estimate of the triangle count $T$.
3: Initialize $A_l, A_m, A_h = 0, S_L, S_M, S_{aux} = \emptyset$
4: **if** $T \geq (m/\epsilon)^{1/2}$ **then**
5:    Set $\rho = (mT)^{1/3}$, $p_1 = \alpha\epsilon^{-2}/\rho$, $p_2 = \min(\beta\epsilon^{-2}\rho/T, 1)$, $p_3 = \gamma\epsilon^{-2}\log n/\rho$.
6: **else**
7:    Set $\rho = m^{1/2}/\epsilon$, $p_1 = 0$, $p_2 = 1$, $p_3 = \gamma\epsilon^{-2}/\rho$. {$\alpha, \beta$, and $\gamma$ are large enough constants.}
8: **while** seeing edges adjacent to $v$ **do**
9:    **for all** $ab \in S_L \cup S_M$ **do**
10:       **if** $a, b \in N(v)$ **then** $C_{ab} = C_{ab} + 1$. {Update the counter for all the sampled light and medium edges}
11:    **for all** $av \in S_{aux}$ **do**
12:       $B_{av} = 1$ {seen $av$ twice}.
13:    **for** each incident edge $vu$ **do**
14:       W.p. $p_3$: $S_{aux} = \{vu\} \cup S_{aux}, B_{vu} = 0$
15:       **if** $O(uv)$ outputs LIGHT **then**
16:          **if** $uv \in S_L$ **then** $A_l = A_l + C_{uv}$. {Finished counting $R_{uv}$}
17:          **else** w.p. $p_1$, $S_L = \{vu\} \cup S_L$
18:       **else** {$uv$ is either MEDIUM or HEAVY}
19:          $\# := |\{z : uz \in S_{aux}, B_{uz} = 1, z \in N(v)\}|$ .
20:          **if** $\# \geq p_3\rho$ **then** {$uv$ is HEAVY}
21:             $A_h = A_h + \#$ .
22:          **else if** $uv \in S_M$ **then**
23:             $A_m = A_m + C_{uv}$. {Finished counting $R_{uv}$}
24:          **else**
24:             With probability $p_2$, $S_M = \{vu\} \cup S_M$ .
25: **return:** $A_l/p_1 + A_m/p_2 + A_h/p_3$.

---

**Remark B.1.** *In the adjacency list model, we assume the oracle can predict whether $R_{xy} \geq \frac{T}{c}$ or not, where $R_{xy}$ is dependent on the node order. This may sometimes be impractical. However, observe that $N_{xy} \geq R_{xy}$ and $\sum N_{xy} = 3T$. Hence, our proof still holds if we assume what the oracle can predict is whether $N_{xy} \geq \frac{T}{c}$, which could be a more practical assumption.*

Recall the following notation from Section 2. We define the edge $xy$ to be *heavy* if $R_{xy} \geq \rho$, *light* if $R_{xy} \leq \frac{T}{\rho}$, and *medium* if $\frac{T}{\rho} < R_{xy} < \rho$. We define $H, M$, and $L$ as the set of heavy, medium, and light edges, respectively. Define $T_L = \sum_{xy \in L} R_{xy}$. We define $T_H$ and $T_M$ similarly. Also, recall that since $\sum_{xy \in E} R_{xy} = T = T_L + T_M + T_H$, it suffices to estimate of these three terms.

**Lemma B.2.** *Let $A$ be an edge set such that for each edge $xy \in A$, $R_{xy} \leq \frac{T}{c}$, for some parameter $c$, and let $S$ be a subset of $A$ such that every element in $A$ is included in $S$ with probability $p = \frac{1}{\tau}$ for $\tau = \epsilon^2 \cdot c/\alpha$, so that $1/\tau = \alpha\epsilon^{-2}\frac{1}{c}$ independently, for a sufficient large constant $\alpha$. Then, with probability at least $9/10$, we have*

$$\tau \sum_{xy \in S} R_{xy} = \sum_{xy \in A} R_{xy} \pm \epsilon T$$

*Proof.* We let $Y = \tau \sum_{i \in S} R_i = \tau \sum_{i \in A} f_i R_i$, where $f_i = 1$ when $i \in S$ and $f_i = 0$ otherwise. It follows that $\mathbb{E}[Y] = T_A$. Then,

$$\mathbf{Var}[Y] = \mathbb{E}[Y^2] - \mathbb{E}[Y]^2$$

$$= \tau^2 \left( \mathbb{E}[\sum_{i \in A} f_i R_i^2 + \sum_{i \neq j} f_i f_j R_i R_j] \right) - T_L^2$$

$$\leq \tau^2 \left( \mathbb{E}[\sum_{i \in A} f_i R_i^2] + \sum_{i,j} \frac{1}{\tau^2} R_i R_j \right) - T_L^2$$

$$= \tau \sum_{i \in A} R_i^2 \ .$$

To bound this, we notice that $R_i \leq \frac{T}{c}$ and $\sum_i R_i = T_A \leq T$, so

$$\tau \sum_{i \in A} R_i^2 \leq \tau \cdot \max_{i \in A} R_i \cdot \sum_{i \in A} R_i \leq \tau \cdot \frac{T}{c} \cdot T = \frac{\tau}{c} T^2 = \frac{1}{\alpha} \epsilon^2 T^2 \ .$$

By Chebyshev's inequality, we have that

$$\Pr[|Y - T_A| > \epsilon T] \leq \frac{\frac{1}{\alpha} \epsilon^2 T^2}{\epsilon^2 T^2} \leq \frac{1}{\alpha} \ .$$

$\square$

The following lemma shows that for the heavy edges, we can use the edges we have sampled to recognize and estimate them.

**Lemma B.3.** *Let $xy$ be an edge in $E$. Assume we sample the edges with rate $p_3 = \beta \epsilon^{-2} \log n / \rho$ in the stream for a sufficiently large constant $\beta$, and consider the set*

$$\# = |xz \text{ has been sampled} : \{x, y, z\} \in \Delta \text{ and } x <_s z <_s y| \ .$$

*Then we have the following: if $R_{xy} \geq \frac{\rho}{2}$, with probability at least $1 - \frac{1}{n^5}$, we have $p_3^{-1} \cdot \# = (1 \pm \epsilon) R_{xy}$. If $R_{xy} < \frac{\rho}{2}$, with probability at least $1 - \frac{1}{n^5}$, we have $p_3^{-1} \cdot \# < \rho$.*

*Proof.* We first consider the case when $R_{xy} \geq \frac{\rho}{2}$.

For each edge $i \in |xz : \{x, y, z\} \in \Delta$ and $x <_s z <_s y|$, let $X_i = 1$ if $i$ has been sampled, or $X_i = 0$ otherwise. We can see the $X_i$ are i.i.d. Bernoulli random variables and $\mathbb{E}[X_i] = p_3$. By a Chernoff bound we have

$$\mathbf{Pr}\left[|\sum_i X_i - p_3 R_{xy}| \geq \epsilon p_3 \cdot R_{xy}\right] \leq 2 \exp\left(-\epsilon^2 p_3 \cdot R_{xy}/3\right) \leq \frac{1}{n^5} \ .$$

Therefore, we have

$$\mathbf{Pr}\left[|p_3^{-1} \cdot \# - R_{xy}| \geq \epsilon R_{xy}\right] = \mathbf{Pr}\left[|p_3^{-1} \cdot \sum_i X_i - R_{xy}| \geq \epsilon R_{xy}\right] \leq \frac{1}{n^5} \ .$$

Alternatively, when $R_{xy} < \frac{\rho}{2}$, we have

$$\mathbf{Pr}\left[|p_3^{-1} \cdot \# - R_{xy}| \geq \frac{1}{2}\rho\right] \leq \frac{1}{n^5} \ ,$$

which means $p_3^{-1} \cdot \# < \rho$ with probability at least $1 - \frac{1}{n^5}$. $\square$

We are now ready to prove Theorem 1.1, restated here for the sake of convenience.

**Theorem 1.1.** *There exists a one-pass algorithm, Algorithm 1, with space complexity[3]* $\widetilde{O}(\min(\epsilon^{-2}m^{2/3}/T^{1/3}, \epsilon^{-1}m^{1/2}))$ *in the adjacency list model that, using a learning-based oracle, returns a* $(1 \pm \epsilon)$*-approximation to the number $T$ of triangles with probability at least[4] $7/10$.*

*Proof.* We first consider the case when $T \geq (m/\epsilon)^{1/2}$.

For each edge $xy \in H$, from Lemma B.3 we have that with probability at least $1 - \frac{1}{n^5}$, $\#/p_3 = (1 \pm \epsilon)R_{xy}$ in Algorithm 1. If $xy \notin H$, then with probability $1 - \frac{1}{n^5}$, $xy$ will not contribute to $A_h$. Taking a union bound, we have that with probability at least $1 - 1/n^3$,

$$A_h/p_3 = (1 \pm \epsilon) \sum_{xy \in H} R_{xy} = (1 \pm \epsilon)T_H .$$

We now turn to deal with the light and medium edges. For a light edge $xy$, it satisfies $R_{xy} \leq T/\rho$ and therefore we can invoke Lemma B.2 with the parameter $c$ set to $\rho$. For a medium edge $xy$, it satisfies $R_{xy} \leq \rho = T/(T/\rho)$, and therefore we can invoke Lemma B.2 with the parameter $c$ set to $T/\rho$. Hence, we have that with probability $4/5$,

$$A_m/p_2 = T_M \pm \epsilon T, A_l/p_1 = T_L \pm \epsilon T .$$

Putting everything together, we have that with probability $7/10$,

$$|A_h/p_3 + A_m/p_2 + A_l/p_1 - T| \leq 3\epsilon T.$$

Now we analyze the space complexity. We note that there are at most $\rho$ medium edges. Hence, the expected total space we use is $O(mp_3 + \rho p_2 + mp_1) = O(\epsilon^{-2}m^{2/3}/T^{1/3} \log n)[5]$.

When $T < (m/\epsilon)^{1/2}$, as described in Section 2, we only have two classes of edges: heavy and light. We will save all the heavy edges and use sub-sampling to estimate $T_L$. The analysis is almost the same. □

## B.1 NOISY ORACLES FOR THE ADJACENCY LIST MODEL

*Proof of Theorem 1.4.* We first consider the case that $T < (m/\epsilon)^{1/2}$. Recall that at this time, our algorithms becomes that we save all the heavy edges $xy$ such that $p(e) \geq \epsilon T/\sqrt{m}$ and sampling the light edges with rate $p_l = \epsilon^{-1}/\sqrt{m}$. We define $\mathcal{L}$ as the set of deemed light edges, $\mathcal{H}$ as the set of deemed heavy edges and $L = |\mathcal{L}|$, $H = |\mathcal{H}|$. Let $\rho = \epsilon T/\sqrt{m}$, then we have the condition that for every edge $e$,

$$1 - K \cdot \frac{\rho}{R_e} \leq Pr[O_\rho(e) = \text{HEAVY}] \leq K \cdot \frac{R_e}{\rho}.$$

We will show that, $(i)$ $\mathbb{E}[H] \leq (K+1)\frac{\sqrt{m}}{\epsilon}$, $(ii)$ $\mathbf{Var}[A_L/p_1] \leq (4K+2)\epsilon^2 T^2$. Under the above conditions we can get that with probability at least $9/10$ we can get a $(1 \pm O(\sqrt{K}) \cdot \epsilon)$-approximation of $T$ by Chebyshev's inequality.

For $(i)$, we divide $\mathcal{H} = H_h \cup H_l$, where the edges in $H_h$ are indeed heavy edges, and the edges in $H_l$ are indeed light edges. Then, it is easy to see that $|H_h| \leq T/\rho$. For every light edges, the probability that it is included in $H_l$ is at most $K \cdot \frac{R_e}{\rho}$, hence we have

$$\mathbb{E}[|H_h|] \leq K \cdot \sum_{e \in \mathcal{H}} (\frac{R_e}{\rho}) \leq K \cdot \frac{T}{\rho} = K\frac{\sqrt{m}}{\epsilon}$$

which implies $\mathbb{E}[H] \leq (K+1)\frac{\sqrt{m}}{\epsilon}$.

For $(ii)$, we also divide $\mathcal{L} = L_l \cup L_h$ similarly. Then we have $\mathbf{Var}[A_L] \leq 2(\mathbf{Var}[X] + \mathbf{Var}[Y])$, where $X = \sum_{e \in L_l, e \text{ has been sampled}} R_e$, $Y = \sum_{e \in L_h, e \text{ has been sampled}} R_e$. Similar to the proof in Lemma B.2, we have

$$\mathbf{Var}[X] \leq p_1 \sum_{e \in \mathcal{L}} R_e^2 \leq p_1 \frac{T}{\rho}\rho^2 = p_1 T\rho.$$

---

[3] We use $\widetilde{O}(f)$ to denote $O(f \cdot \text{polylog}(f))$.

[4] The success probability can be $1 - \delta$ by running $\log(1/\delta)$ copies of the algorithm and taking the median.

[5] Using Markov's inequality, we have that with probability at least $9/10$, the total space we use is less than 10 times the expected total space.

Now we bound $\mathbf{Var}[Y]$. For every heavy edge we have that $\mathbf{Pr}[e \in L_h] \leq K \cdot \frac{\rho}{R_e}$. Then, condition on the oracle $O_\rho$, we have

$$\mathbb{E}[\mathbf{Var}[Y|O_\rho]] \leq p_1 K \sum_{e \in \mathcal{H}} \frac{\rho}{R_e} R_e^2 \leq p_1 K \sum_{e \in \mathcal{H}} \rho R_e \leq p_1 K \rho T .$$

Also, we have

$$\mathbf{Var}[\mathbb{E}[Y|O_\rho]] < p_1 K \sum_{e \in \mathcal{H}} \frac{\rho}{R_e} R_e^2 \leq p_1 K \rho T .$$

From $\mathbf{Var}[Y] = \mathbb{E}[\mathbf{Var}[Y|O_\rho]] + \mathbf{Var}[\mathbb{E}[Y|O_\rho]]$ we know $\mathbf{Var}[Y] \leq 2p_1 K \rho T$, which means $\mathbf{Var}[A_l/p_1] \leq \frac{1}{p_1}(4K+2)\rho T = (4K+2)\epsilon^2 T^2$.

When $T \geq (m/\epsilon)^{1/2}$, recall that the oracle $O_\rho$ only needs to predict the edges that are medium edges or light edges. So, we can use the same way to bound the expected space for the deemed medium edges and the variance of the deemed light edges. □

## B.2 A Value Prediction Oracle for the Adjacency List Model

In this section, we consider two types of value-prediction oracles, and variants of Algorithm 1 that take advantage of these oracles. Recall that given an edge $xy$, $R_{xy} = |z : \{x, y, z\} \in \Delta$ and $x <_s z <_s y|$ .

**Definition B.4.** *A value-prediction oracle with parameter $(\alpha, \beta)$ is an oracle that, for any edge $e$, outputs a value $p(e)$ for which $\mathbb{E}[p(e)] \leq \alpha R_e + \beta$, and $\mathbf{Pr}[p(e) < \frac{R_e}{\lambda} - \beta] \leq K e^{-\lambda}$ for some constant $K$ and any $\lambda \geq 1$.*

Our algorithm is based on the following stability property of exponential random variables (see e.g. Andoni (2017))

**Lemma B.5.** *Let $x_1, x_2, ..., x_n > 0$ be real numbers, and let $u_1, u_2, ...u_n$ be i.i.d. standard exponential random variables with parameter $\lambda = 1$. Then we have that the random variable $\max(\frac{x_1}{u_1}, ..., \frac{x_n}{u_n}) \sim \frac{x}{u}$, where $x = \sum_i x_i$ and $u \sim \mathrm{Exp}(1)$.*

**Lemma B.6.** *Let $r = O(\ln(1/\delta)/\epsilon^2)$ and $\epsilon \leq 1/3$. If we take $r$ independent samples $X_1, X_2, ..., X_r$ from $\mathrm{Exp}(1)$ and let $X = \mathrm{median}(X_1, X_2, ..., X_r)$, then with probability $1 - \delta$, $X \in [\ln 2(1 - \epsilon), \ln 2(1 + 5\epsilon)]$.*

*Proof.* We first prove the following statement: with probability $1 - \delta$, $F(X) \in [1/2 - \epsilon, 1/2 + \epsilon]$, where $F$ is the cumulative density function of a standard exponential random variable.

Consider the case when $F(X) \leq 1/2 - \epsilon$. We use the indicator variable $Z_i$ where $Z_i = 1$ if $F(X_i) \leq 1/2 - \epsilon$ or $Z_i = 0$ otherwise. Notice that when $F(X) \leq 1/2 - \epsilon$, at least half of the $Z_i$ are 1, so $Z = \sum_i Z_i \geq r/2$. On the other hand, we have $\mathbb{E}[Z] = r/2 - r\epsilon$, by a Chernoff bound. We thus have

$$\mathbf{Pr}[|Z - \mathbb{E}[Z]| \geq \epsilon r] \leq 2e^{-\epsilon^2 r/3} \leq \delta/2 .$$

Therefore, we have with probability at most $\delta/2$, $F(X) \leq 1/2 - \epsilon$. Similarly, we have that with probability at most $\delta/2$, $F(X) \geq 1/2 + \epsilon$. Taking a union bound, we have that with probability at least $1 - \delta$, $F(X) \in [1/2 - \epsilon, 1/2 + \epsilon]$.

Now, condition on $F(X) \in [1/2 - \epsilon, 1/2 + \epsilon]$. Note that $F^{-1}(x) = -\ln(1 - x)$, so if we consider the two endpoints of $F(X)$, we have

$$F^{-1}(1/2 - \epsilon) = -\ln(1/2 + \epsilon) \geq (1 - \epsilon)\ln 2,$$

and

$$F^{-1}(1/2 + \epsilon) = -\ln(1/2 - \epsilon) \leq (1 + 5\epsilon)\ln 2.$$

which indicates that $X \in [\ln 2(1 - \epsilon), \ln 2(1 + 5\epsilon)]$. □

The algorithm utilizing this oracle is shown in Algorithm 2. As described in Section 2, here we runs $O(\frac{1}{\epsilon^2})$ copies of the subroutine and takes a medium of them. For each subroutine, we aim to output the maximum value $\max_e R_e/u_e$. We will show that with high probability for each subroutine,

this maximum will be at least $T/\log(K/\epsilon)$. Hence, we only need to save the edge $e$, such that $(R_e + \beta)/u_e$ is larger than $T/\log(K/\epsilon)$. However, one issue here is we need the information of $T$. We can remove this assumption when $\beta = 0$ using the following way: we will show that with high probability, the total number of edges we save is less than $H = O(\frac{1}{\epsilon^2}\log^2(K/\epsilon)(\alpha + m\beta/T))$, so every time when a new edge comes, we can search the minimum value $M$ such that the total number of edges $e$ that in at least one subroutine $(p(e) + \beta)/u_e > M$ is less than $H$(we count an edge multiple times if it occurs in multiple subroutines), then use $M$ as the threshold. We can see $M$ will increase as the new edge arriving and it will be always less than $H$.

---

**Algorithm 2** Counting Triangles in the Adjacency List Model with a Value Prediction Oracle

1: **Input:** Adjacency list edge stream and an value prediction oracle with parameter$(\alpha, \beta)$.
2: **Output:** An estimate of the number $T$ of triangles.
3: Initialize $X \leftarrow \emptyset$ and $H = O(\frac{1}{\epsilon^2}\log^2(K/\epsilon)(\alpha + m\beta/T))$, and let $c$ be some large constant.
4: **for** $i = 0$ **to** $c\epsilon^{-2}$ **do** {Initialize sets for $c\epsilon^{-2}$ copies of samples, where $S_i$ stores edges, $Q_i$ stores all the exponential random variables for each sampled edge, and $A_i$ stores the current maximum of each sample}
5:   $S^i \leftarrow \emptyset, Q^i \leftarrow \emptyset$, and $A^i \leftarrow 0$ .
6: **while** seeing edges adjacent to $y$ in the stream **do**
7:   **for** $i = 0$ **to** $c\epsilon^{-2}$ **do** {Update the counters for each sample}
8:    For all $ab \in S^i$: if $a, b \in N(y)$ then $C^i_{ab} \leftarrow C^i_{ab} + 1$
9:   **for** each incident edge $yx$ **do**
10:    **for** $i = 0$ **to** $c\epsilon^{-2}$ **do**
11:     **if** $xy \in S^i$ **then** {Finished counting $C_{xy}$, update the current maximum}
12:      $A^i \leftarrow \max(A^i, C^i_{xy}/u^i_{xy})$ .
13:     **else** {Put $yx$ into the sample sets temporarily}
14:      Let $\hat{R}_{yx}$ be the predicted value of $R_{yx}$.
15:      $\hat{R}_{yx} \leftarrow \hat{R}_{yx} + \beta$.
16:      Generate $u^i_{yx} \sim \text{Exp}(1)$. Set $Q^i \leftarrow Q^i \cup \{u^i_{yx}\}, S^i \leftarrow S^i \cup \{yx\}$.
17:   Set $M$ to be the minimal integer such that $\sum_i s_i \leq H$, where $s_i = |\{e \in E \mid \hat{R}_e/u^i_e \geq M\}|$.

18:   **for** $i = 0$ **to** $c\epsilon^{-2}$ **do** {Adjust the samples to be within the space limit}
19:    Let $Q^i \leftarrow Q^i \setminus \{u^i_{ab}\}, S^i \leftarrow S^i \setminus \{ab\}$ if $\hat{R}_{ab}/u^i_{ab} < M$ for all $ab \in E$.
20: **for** $i = 0$ **to** $c\epsilon^{-2}$ **do**
21:   $X \leftarrow X \cup \{A^i\}$
22: **return:** $\ln 2 \cdot \text{median}(X)$ .

---

We are now ready to prove Theorem 1.5. We first recall the theorem.

**Theorem 1.5.** *Given an oracle with parameters $(\alpha, \beta)$, there exists a one-pass algorithm, Algorithm 2, with space complexity $O(\epsilon^{-2}\log^2(K/\epsilon)(\alpha + m\beta/T))$ in the adjacency list model that returns a $(1 \pm \epsilon)$-approximation to the number of triangles $T$ with probability at least $7/10$.*

*Proof.* It is easy to see that for the $i$-th subroutine, the value $f(i) = \max_e R_e/u^i_e$ is the sample from the distribution $T/u$, where $u \sim \text{Exp}(1)$. And if the edge $e_i$, for which the true value $R_{e_i}/u^i_{e_i}$ is the maximum value among $E$, is included in $S^i$, then the output of the $i$-th subroutine will be a sample from $T/u$. Intuitively, the prediction value $p(e)$ will help us find this edge.

We will first show that with probability at least $9/10$ the following events will happen:

- $(i)$ $f(i) \geq c_1 T/\log(1/\epsilon)$ for all $i$, where $c_1$ is a constant.

- $(ii)$ Let $s_i$ be the number of the edge $e$ in the $i$-th subroutine such that $(p(e) + \beta)/u^i_e \geq c_1^2 T/\log^2(K/\epsilon)$, then $\sum_i s_i \leq c_2(\frac{1}{\epsilon^2}\log^2(K/\epsilon)(\alpha + m\beta/T))$ for some constant $c_2$.

- $(iii)$ $p(e_i) + \beta \geq c_1 f(i)/\log(K/\epsilon)$ for all $i$.

For $(i)$, recall that $f(i) \sim T/u$, where $u \sim \text{Exp}(1)$. Hence, we have

$$\mathbf{Pr}[f(i) < c_1 T / \log(1/\epsilon)] = e^{-\log(1/\epsilon)/c_1} \leq \frac{1}{100c} \frac{1}{\epsilon^2} .$$

Taking an union bound we get that with probability at least $99/100$, $(i)$ will happen. For each edge $e$, we have

$$\mathbf{Pr}[p(e) + \beta < R_e/\lambda] < Ke^{-\lambda},$$

similarly we can get that with probability at least $99/100$, $(iii)$ will happen.

For $(ii)$, we first bound the expectation of $s_i$. For each edge $e$, we have

$$\mathbf{Pr}\left[(p(e) + \beta)/u_e^i \geq c_1^2 T / \log^2(K/\epsilon)\right] = \mathbf{Pr}[u_e \leq c_1^2(p(e) + \beta) \log^2(K/\epsilon)/T]$$
$$\leq c_1^2(p(e) + \beta) \log^2(K/\epsilon)/T$$
$$\leq c_1^2(\alpha R_e + 2\beta) \log^2(K/\epsilon)/T .$$

Hence, we have

$$\mathbb{E}[s_i] = \sum_e c_1^2 \frac{(\alpha R_e + 2\beta) \log^2(K/\epsilon)}{T} = O(\log^2(K/\epsilon)(\alpha + m\beta/T)).$$

Using Markov's inequality, we get that with probability at least $99/100$, $(ii)$ will happen. Finally, taking a union bound, we can get with probability at least $9/10$, all the three events will happen.

Now, conditioned on the above three events, we will show that all the subroutine $i$ will output $f(i)$. For the subroutine $i$, from $(ii)$ we know that $M$ will be always smaller than $O(T/\log^2(K/\epsilon))$, and from $(i)$ and $(iii)$ we get that $(p(e_i) + \beta)/u_{e_i}^i \geq \Omega(T/\log^2(K/\epsilon))$. Hence, the edge $e_i$ will be saved in the subroutine $i$. From Lemma B.6 we get that with probability at least $7/10$, we can finally get a $(1 \pm \epsilon)$-approximation of $T$. $\qquad \square$

We note that when $\beta = 0$, the space complexity will become $O(\frac{1}{\epsilon^2} \log^2(K/\epsilon)\alpha)$, and in this case we will not need a lower bound of $T$.

We will also consider the following noise model.

**Definition B.7.** *A value-prediction oracle with parameter $(\alpha, \beta)$ is an oracle that, for any edge $e$, outputs a value $p(e)$ for which $\mathbf{Pr}[p(e) > \lambda\alpha R_e + \beta] \leq \frac{K}{\lambda}$, and $\mathbf{Pr}[p(e) < \frac{R_e}{\lambda} - \beta] \leq \frac{K}{\lambda}$ for some constant $K$ and any $\lambda \geq 1$.*

The algorithm for this oracle is shown in Algorithm 3.

We now state our theorem.

**Theorem B.1.** *Given an oracle with parameter $(\alpha, \beta)$, there exists a one-pass algorithm, Algorithm 3, with space complexity $O(\frac{K}{\epsilon^2}(\alpha(\log n)^3 \log\log n + m\beta/T))$ in the adjacency list model that returns a $(1 \pm \sqrt{K} \cdot \epsilon)$-approximation to $T$ with probability at least $7/10$.*

*Proof.* Define $q(e) = p(e) + \beta$, it is easy to see that from the condition we can get that

$$\mathbf{Pr}[q(e) > 2\lambda\alpha R_e] \leq K\frac{1}{\lambda} \text{ when } R_e > 2\beta, \mathbf{Pr}\left[q(e) < \frac{R_e}{\lambda}\right] \leq K\frac{1}{\lambda}$$

We define the set $E_i$ such that $e \in E_i$ if and only if $q(e) \in I_i$. We can see $\mathbb{E}[A_i/p_i] = \sum_{e \in E_i} R_e$. Now, we bound $\mathbf{Var}[A_i/p_i]$ when $i \geq 1$.

Let $X = \sum_{e \in E_i, R_e \leq 2^{i+1}\beta} R_e$ and $Y = \sum_{e \in E_i, R_e > 2^{i+1}\beta} R_e$, then we have $\mathbf{Var}[A_i] \leq 2(\mathbf{Var}[X] + \mathbf{Var}[Y])$.

Similar to the proof in Lemma B.2, we can get that

$$\mathbf{Var}[X] \leq p_i \sum_{e \in E_i, R_e \leq 2^{i+1}\beta} R_e^2 \leq p_i(2^{i+1}\beta)^2 \frac{T}{2^{i+1}\beta} = p_i 2^{i+1}\beta T.$$

---

**Algorithm 3** Counting Triangles in the Adjacency List Model with a Value Prediction Oracle

1: **Input:** Adjacency list edge stream and an value prediction oracle with parameter$(\alpha, \beta)$.
2: **Output:** An estimate of the number $T$ of triangles.
3: Initialize $S_i^j \leftarrow \emptyset$ and $A_i^j \leftarrow 0$ where $i = O(\log n)$ and $j = O(\log \log n)$. $p_0 = c\epsilon^{-2}2^i\beta/T$ and $p_i = c\epsilon^{-2}2^i\beta(\log n)^2/T$ for some constant $c$. Define $I_0 = [0, 2\beta)$ and $I_i = [2^i\beta, 2^{i+1}\beta)$, $H \leftarrow 0$.
4: **while** seeing edges adjacent to $y$ in the stream **do**
5:     For all $ab \in S_i^j$: if $a, b \in N(y)$ then $C_{ab} \leftarrow C_{ab} + 1$
6:     **for** each incident edge $yx$ **do**
7:         **if** $xy \in S_i^j$ **then**
8:             $A_i^j \leftarrow A_i^j + C_{yx}$ .
9:         **else**
10:             Let $\hat{R}_{yx}$ be the predicted value of $R_{yx}$.
11:             Search $i$ such that $(\hat{R}_{yx} + \beta) \in I_i$
12:             For each $j$ let $S_i^j \leftarrow S_i^j \cup \{yx\}$ with probability $p_i$.
13: **for** $i = 0$ to $O(\log n)$ **do**
14:     $H \leftarrow H + \text{median}(A_i^j)/p_i$
15: **return:** $H$ .

---

For $\mathbf{Var}[Y]$, recall that for an edge $e$ such that $R_e \geq 2^{i+1}\beta$, we have $\mathbf{Pr}[e \in E_i] \leq K\frac{q(e)}{R_e} \leq K\frac{2^{i+1}\beta}{R_e}$. Then, condition on the oracle $O_\theta$, we have

$$\mathbb{E}[\mathbf{Var}[Y|O_\theta]] \leq p_i \sum_{R_e \geq 2^{i+1}\beta} K\frac{2^{i+1}\beta}{R_e}R_e^2 = p_i K 2^{i+1}\beta \sum_{R_e \geq 2^{i+1}\beta} R_e = p_i K 2^{i+1}\beta T.$$

And

$$\mathbf{Var}[\mathbb{E}[Y|O_\theta]] < p_i \sum_{R_e \geq 2^{i+1}\beta} K\frac{2^{i+1}\beta}{R_e}R_e^2 = p_i K 2^{i+1}\beta T.$$

From $\mathbf{Var}[Y] = \mathbb{E}[\mathbf{Var}[Y|O_\theta]] + \mathbf{Var}[\mathbb{E}[Y|O_\theta]]$ we get that $\mathbf{Var}[Y] \leq 2p_i K 2^{i+1}\beta T$, from which we can get that $\mathbf{Var}[A_i/p_i] = \frac{1}{p_i^2}\mathbf{Var}[A_i] = O(K(\epsilon/\log n)^2 T^2)$. Using Chebyshev's inequality and taking a median over $O(\log \log n)$ independent trials, we can get $X_i = \text{median}(A_i^j)$ satisfies $|X_i - \sum_{e \in E_i} R_e| \leq \sqrt{K}(\epsilon/\log n)T$ with probability $1 - O(1/\log n)$. A similar argument also holds for $I_0$, which means that after taking a union bound, with probability $9/10$, the output of the algorithm 3 is a $(1 \pm \sqrt{K} \cdot \epsilon)$- approximation of $T$.

Now we analyze the space complexity of Algorithm 3.

For $i \geq 1$, we have

$$\mathbb{E}[|E_i|] = \mathbb{E}\left[\sum_{R_e \leq 2^{i-1}\beta/\alpha} [e \in E_i] + \sum_{R_e > 2^{i-1}\beta/\alpha} [e \in E_i]\right] \leq \mathbb{E}\left[\sum_{R_e \leq 2^{i-1}\beta/\alpha} [e \in E_i]\right] + \frac{T\alpha}{2^{i-1}\beta}$$

$$\leq \sum_{R_e \leq 2^{i-1}\beta/\alpha} 2K\frac{R_e}{p(e)} + \frac{T\alpha}{2^{i-1}\beta} \leq \sum_{R_e \leq 2^{i-1}\beta/\alpha} K\frac{R_e}{2^{i-1}\beta} + \frac{T\alpha}{2^{i-1}\beta} \leq (K+1)\frac{T\alpha}{2^{i-1}\beta},$$

and $|E_0| \leq m$, hence we have that the total expectation of the space is $\sum_i p_i \mathbb{E}[|E_i|] = O(\frac{K}{\epsilon^2}(\alpha(\log n)^3 \log \log n + m\beta/T))$. $\qquad\square$

## C  OMITTED PROOFS FROM SECTION 3

We first present Algorithm 4, and then continue to prove Theorem 1.2. In the following algorithm, for two sets $A, B$ of edges, we let $w_{A,B}$ represent the set of pairs of edges $(u, v)$, where $u \in A, v \in B$, and $u, v$ share a common vertex.

---

**Algorithm 4** Counting Triangles in the Arbitrary Order Model

---

1: **Input:** Arbitrary order edge stream and an oracle $O_\rho$ that outputs HEAVY if $N_{xy} > \rho$ and LIGHT otherwise.
2: **Output:** An estimate of the triangle count $T$.
3: Initialize $\rho = \max\left\{\frac{\epsilon T}{\sqrt{m}}, 1\right\}$, $p = C\max\left\{\frac{1}{\epsilon\sqrt{T}}, \frac{\rho}{\epsilon^2 \cdot T}\right\}$ for a constant $C$.
4: Initialize $\ell_1, \ell_2, \ell_3 = 0$, and $S_L, H = \emptyset$.
5: **for** every arriving edge $e$ in the stream **do**
6:     **if** the oracle $O_\rho$ outputs HEAVY **then**
7:         Add $e$ to $H$
8:     **else**
9:         Add $e$ to $S_L$ with probability $p$.
10:     **for** each $w \in w_{S_L, S_L}$ **do**
11:         **if** $(e, w)$ is a triangle **then**
12:             Increment $\ell_1 = \ell_1 + 1$.
13:     **for** each $w \in w_{S_L, H}$ **do**
14:         **if** $(e, w)$ is a triangle **then**
15:             Increment $\ell_2 = \ell_2 + 1$.
16:     **for** each $w \in w_{H, H}$ **do**
17:         **if** $(e, w)$ is a triangle **then**
18:             Increment $\ell_3 = \ell_3 + 1$.
19: **return:** $\ell = \ell_1/p^2 + \ell_2/p + \ell_3$.

---

## C.1   Proof of Theorem 1.2

Recall that we first assume that Algorithm 4 is given access to a perfect oracle $O_\rho$. That is, for every edge $e \in E$,

$$O_\rho(e) = \begin{cases} \text{LIGHT} & \text{if } T_e \leq \rho \\ \text{HEAVY} & \text{if } T_e > \rho, \end{cases}$$

where $T_e$ as the number of triangles incident to the edge $e$.

*Theorem 1.2.* First, we assume that $T \geq \epsilon^{-2}$. If not, our algorithm can just store all of the edges, since $m \leq \epsilon^{-1} \cdot m/\sqrt{T}$.

For each integer $i \geq 1$, let $\mathcal{E}(i)$ denote the set of edges that are part of exactly $i$ triangles, and let $E_i = |\mathcal{E}(i)|$. Since each triangle has 3 edges, we have that $\sum_{i \geq 1} i \cdot E(i) = 3T$.

Let $\mathcal{T}_1$ denote the set of triangles whose first two edges in the stream are light (according to $O_\rho$). For every triangle $t_i$ in $\mathcal{T}_1$, let $\chi_i$ denote the indicator of the event that the first two edges of $t_i$ are sampled to $S_L$, and let $\ell_1 = \sum_i \chi_i$. Since each $\chi_i$ is 1 with probability $p^2$, $\mathbf{Ex}[\chi_i] = p^2$, and $\mathbf{Var}[\chi_i] = p^2 - p^4 \leq p^2$. For any two variables $\chi_i, \chi_j$, they must be uncorrelated unless the triangles $t_i, t_j$ share a light edge that is among the first two edges of $t_i$ and among the first two edges of $t_j$. Moreover, in this case, $\mathbf{Cov}[\chi_i, \chi_j] \leq \mathbf{Ex}[\chi_i\chi_j] = \mathbb{P}(\chi_i = \chi_j = 1)$. This event only happens if the first two edges of both $t_i$ and $t_j$ are sampled in $S_L$, but due to the shared edge, this comprises 3 edges in total, so $\mathbb{P}(\chi_i = \chi_j = 1) = p^3$. Hence, if we define $t_i^{12}$ as the first two edges of $t_i$ in the stream for each triangle $t_i$,

$$\mathbf{Ex}[\ell_1] = p^2|\mathcal{T}_1|,$$

and

$$\mathbf{Var}[\ell_1] \leq \sum_{t_i \in \mathcal{T}_1} p^2 + \sum_{\substack{t_i, t_j \in \mathcal{T}_1 \\ t_i^{12} \cap t_j^{12} \neq \emptyset}} p^3 \ \leq \ p^2|\mathcal{T}_1| + p^3 \sum_{T_e \leq \rho} T_e^2$$

$$= p^2|\mathcal{T}_1| + p^3 \cdot \sum_{t=1}^{\rho} t^2 \cdot E(t) \ \leq \ (p^2 + 3p^3 \cdot \rho) \cdot T \ .$$

The first line follows by adding the covariance terms; the second line follows since each light edge $e$ has at most $T_e^2$ pairs $(t_i, t_j)$ intersecting at $e$; the third line follows by summing over $t = T_e$, ranging from 1 to $\rho$, instead over $e$; and the last line follows since $\sum_{t \geq 1} t \cdot E(t) = 3T$.

Now, let $\mathcal{T}_2$ denote the set of triangles whose first two edges in the stream are light and heavy (in either order). For every triangle $t = (e_1, e_2, e_3)$ in $\mathcal{T}_2$, $e_1, e_2$ are sampled to $S_L \cup H$ with probability $p$. Also all other triangles have no chance to contribute to $\ell_2$. Hence, $\mathbf{Ex}[\ell_2] = p \cdot \mathcal{T}_2$. Also, two triangles $t_i, t_j \in \mathcal{T}_2$ that intersect on an edge $e$ have $\mathbf{Cov}(\chi_i, \chi_j) \neq 0$ only if $e$ is light and $e \in t_i^{12}, t_j^{12}$. In this case.

$$\mathbf{Cov}(\chi_i, \chi_j) \leq \mathbf{Ex}[\chi_i \chi_j] \leq p,$$

since if $e$ is added to $S_L$ (which happens with probability $p$) then $\chi_i = \chi_j = 1$ as the other edges in $t_i^{12}, t_j^{12}$ are heavy and are added to $H$ with probability 1. Therefore,

$$\mathbf{Var}[\ell_2] = \sum_{t_i \in \mathcal{T}_2} p + \sum_{\substack{t_i, t_j \in \mathcal{T}_2 \\ t_i^{12} \cap t_j^{12} = e, \, O_\rho(e) = \text{LIGHT}}} p \leq (p + 3p \cdot \rho) \cdot T,$$

by the same calculations as was done to compute $\mathbf{Var}[\ell_1]$.

Finally, let $\mathcal{T}_3$ denote the set of triangles whose first two edges in the stream are heavy. Then $\ell_3 = |\mathcal{T}_3|$.

Now, using the well known fact that $\mathbf{Var}[X+Y] \leq 2(\mathbf{Var}[X]+\mathbf{Var}[Y])$ for any (possibly correlated) random variables $X, Y$, we have that

$$\mathbf{Var}[p^{-2}\ell_1 + p^{-1}\ell_2 + \ell_3] \leq 2 \left( p^{-4}(p^2 + 3p^3\rho)T + p^{-2}(p + 3p\rho)T \right)$$
$$\leq 4T(p^{-2} + 3p^{-1}\rho),$$

since $\ell_3$ has no variance. Moreover, $\mathbf{Ex}[p^{-2}\ell_1 + p^{-1}\ell_2 + \ell_3] = |\mathcal{T}_1| + |\mathcal{T}_2| + |\mathcal{T}_3| = T$. So, by Chebyshev's inequality, since $\ell = \ell_1/p^2 + \ell_2/p + \ell_3$,

$$\Pr[|\ell - T| > \epsilon T] < \frac{\mathbf{Var}[\ell]}{\epsilon^2 \cdot T^2} < \frac{4(p^{-2} + 3p^{-1}\rho)}{\epsilon^2 \cdot T} .$$

Therefore, setting $p = C \cdot \max \left\{ 1/(\epsilon \cdot \sqrt{T}), \rho/(\epsilon^2 \cdot T) \right\}$ for some fixed constant $C$ implies that, with probability at least $2/3$, $\ell$ is a $(1 \pm \epsilon)$-multiplicative estimate of $T$.

Furthermore, the expected space complexity is $O(mp + H) = O(mp + T/\rho)$. Setting $\rho = \max\{\epsilon T/\sqrt{m}, 1\}$ implies that the space complexity is $O(\epsilon^{-1}m/\sqrt{T} + \epsilon^{-2}m/T + \epsilon^{-1}\sqrt{m} \cdot T/T) = O(\epsilon^{-1}m/\sqrt{T} + \epsilon^{-1}\sqrt{m} \cdot T/T)$, since we are assuming that $T \geq \epsilon^{-2}$. Hence, assuming that $T$ is a constant factor approximation of $T$, the space complexity is $O(\epsilon^{-1}m/\sqrt{T} + \epsilon^{-1}\sqrt{m})$. $\qquad\square$

## C.2 Proof of Theorem 1.2 for the K-noisy oracle

In this section, we prove Theorem 1.2 for the case that the given oracle is a $K$-noisy oracle, as defined in Definition 1.1. That is, we prove the following:

**Theorem C.1.** *Suppose that the oracle in Algorithm 4 is a $K$-noisy oracle as defined in Definition 1.1 for a fixed constant $K$. Then with probability $2/3$, Algorithm 4 returns $\ell \in (1 \pm \epsilon)T$, and uses space at most $O\left(\epsilon^{-1}\left(m/\sqrt{T} + \sqrt{m}\right)\right)$.*

Recall that in Definition 1.1, we defined a $K$-noisy oracle if the following holds. For every edge $e$, $1 - K \cdot \frac{\rho}{N_e} \leq \Pr[O_\rho(e) = \text{HEAVY}] \leq K \cdot \frac{N_e}{\rho}$. We first visualize this model. To visualize this error model, in Figure 3 we have plotted $N_e$ versus the range $\left[1 - K \cdot \frac{\rho}{N_e}, K \cdot \frac{N_e}{\rho}\right]$ for $K = 2$. We set $N_e$ to vary on a logarithmic scale for clarity. For example, if $N_e$ exceeds the threshold $\rho$ by a factor of 2, there is no restriction on the oracle output, whereas if $N_e$ exceeds $\rho$ by a factor of 4, then the oracle must classify the edge as heavy with probability at least $0.5$. In contrast, the blue piece-wise line shows the probability $\Pr[O_\rho(e) = \text{HEAVY}]$ if the oracle $O_\rho$ is a perfect oracle.

*Proof of Theorem C.1.* We follow the proof of Theorem 4.1 from the main paper. Let $\mathcal{T}_1$ be the set of triangles such that their first two edges in the stream are light according to the oracle. Let $\mathcal{T}_2$ be the set of triangles such that their first two edges in the stream are determined light and heavy according to the oracle. Finally, let $\mathcal{T}_3$ be the set of triangles for which their first two edges are

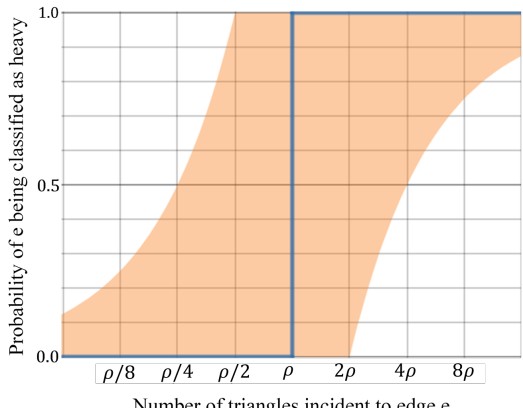

Figure 3: Plot of $N_e$, the number of triangles containing edge $e$, versus the allowed oracle probability range $\Pr[O_\rho(e) = \text{HEAVY}]$, shaded in orange. The blue piece-wise line shows the probability $\Pr[O_\rho(e) = \text{HEAVY}]$ if the oracle $O_\rho$ is a perfect oracle.

determined heavy by the oracle. Furthermore, we define $\chi_i$ for each triangle $t_i$, $t(e)$ for each edge $e$, and $\mathcal{E}(i), E_i$ for each $i \geq 1$ as done in Theorem 4.1. Finally, we define $\mathcal{L}(i)$ as the set of deemed light edges that are part of exactly $i$ triangles, and $L(i) = |\mathcal{L}(i)|$.

First, note that the expectation (over the randomness of $O_\rho$) of $L(i)$ is at most $E(i) \cdot K \cdot \frac{\rho}{i}$, since our assumption on $\Pr[O_\rho(e) = \text{heavy}]$ tells us that $\Pr[O_\rho(e) = \text{light}] \leq K \cdot \frac{\rho}{i}$ if $t(e) = i$. Therefore, by the same computation as in Theorem 4.1, we have that, conditioning on the oracle, $\mathbf{Ex}[\ell_1|O_\rho] = p^2|\mathcal{T}_1|$ and

$$
\begin{aligned}
\mathbf{Var}[\ell_1|O_\rho] &= \sum_{t_i \in \mathcal{T}_1} p^2 + \sum_{\substack{t_i, t_j \in \mathcal{T}_1 \\ t_i^{12} \cap t_j^{12} \neq \emptyset}} p^3 \\
&\leq p^2|\mathcal{T}_1| + p^3 \sum_{e \text{ light}} t(e)^2 \\
&= p^2|\mathcal{T}_1| + p^3 \cdot \sum_{t \geq 1} t^2 \cdot L(t)
\end{aligned}
$$

But since $\mathbf{Ex}_{O_\rho}[L(t)] \leq \frac{K \cdot \rho}{t} \cdot E(t)$ for all $t$, we have that

$$
\begin{aligned}
\mathbf{Ex}_{O_\rho}[\mathbf{Var}[\ell_1|O_\rho]] &\leq \mathbf{Ex}_{O_\rho}\left[p^2 \cdot |\mathcal{T}_1| + p^3 \cdot \sum_{t \geq 1} t^2 L(t)\right] \\
&= p^2|\mathcal{T}_1| + K\rho \cdot p^3 \sum_{t \geq 1} t \cdot E(t) \\
&\leq (p^2 + 3Kp^3 \cdot \rho) \cdot T.
\end{aligned}
$$

A similar analysis to the above shows that $\mathbf{Ex}[\ell_2] = p \cdot |\mathcal{T}_2|$ and that

$$
\begin{aligned}
\mathbf{Ex}_{O_\rho}[\mathbf{Var}[\ell_2|O_\rho]] &\leq \mathbf{Ex}_{O_\rho}\left[p \cdot |\mathcal{T}_2| + p \cdot \sum_{t \geq 1} t^2 L(t)\right] \\
&= p|\mathcal{T}_2| + K\rho \cdot p \sum_{t \geq 1} t \cdot E(t) \\
&\leq (p + 3Kp \cdot \rho) \cdot T.
\end{aligned}
$$

Hence, as in Theorem 4.1, we have $\mathbf{Ex}[\ell|O_\rho] = T$ and $\mathbf{Ex}_{O_\rho}[\mathbf{Var}[\ell|O_\rho]] \leq 4T \cdot (p^{-2} + 3Kp^{-1}\rho)$. Thus, $\mathbf{Ex}[\ell] = \mathbf{Ex}[\mathbf{Ex}[\ell|O_\rho]] = T$ and $\mathbf{Var}[\ell] = \mathbf{Ex}_{O_\rho}[\mathbf{Var}[\ell|O_\rho]] + \mathbf{Var}_{O_\rho}[\mathbf{Ex}[\ell|O_\rho]] \leq 4T \cdot (p^{-2} + 3Kp^{-1}\rho)$ by the laws of total expectation/variance. Therefore, since $K$ is a constant, the variance is

the same as in Theorem 4.1, up to a constant. Therefore, we still have that $\ell \in (1 \pm O(\epsilon)) \cdot T$ with probability at least $2/3$.

Finally, the expected space complexity is bounded by

$$mp + \sum_{e \in m} Pr[O_\rho(e) = \text{heavy}] \leq mp + \sum_{t \geq 1} E(t) \cdot K \cdot \frac{t}{\rho}$$

$$= mp + \frac{K}{\rho} \cdot \sum_{t \geq 1} E(t) \cdot t$$

$$= O(mp + T/\rho),$$

since $\sum_{t \geq 1} E(t) \cdot t = 3T$ and $K = O(1)$. Hence, setting $\rho$ and $p$ as in the proof of Theorem 4.1 implies that the returned value is a $(1 \pm \epsilon)$-approximation of $T$, and the the space complexity is $O(\epsilon^{-1}(m/\sqrt{T} + \sqrt{m}))$ as before. $\qquad\square$

## C.3 LOWER BOUND

In this section, we prove a lower bound for algorithms in the arbitrary order model that have access to a perfect heavy edge oracle. Here heavy means $N_{uv} \geq T/c$ for a pre-determined threshold $c$, and we assume $c = o(m)$ and $T/c > 1$, as otherwise the threshold will be too small or too close to the average to give an accurate prediction. The following theorem shows an $\Omega(\min(m/\sqrt{T}, m^{3/2}/T))$ lower bound even with such an oracle.

**Theorem C.2.** *Suppose that the threshold of the oracle $c = O(m^q)$, where $0 \leq q < 1$. Then for any $T$ and $m$, $T \leq m$, there exists $m' = \Theta(m)$ and $T' = \Theta(T)$ such that any one-pass arbitrary order streaming algorithm that distinguishes between $m'$ edge graphs with $0$ and $T'$ triangles with probability at least $2/3$ requires $\Omega(m/\sqrt{T})$ space. For any $T$ and $m$, $T = \Omega(m^{1+\delta})$ where $\max(0, q - \frac{1}{2}) \leq \delta < \frac{1}{2}$, there exists $m' = \Theta(m)$ and $T' = \Theta(T)$ such that any one-pass arbitrary order streaming algorithm that distinguishes between $m'$ edge graphs with $0$ and $T'$ triangles with probability at least $2/3$ requires $\Omega(m^{3/2}/T)$ space.*

When $T \leq m$, we consider the hubs graph mentioned in Kallaugher & Price (2017).

**Definition C.3** (Hubs Graph, Kallaugher & Price (2017))**.** *The hubs graph $H_{r,d}$ consists of a single vertex $v$ with $2rd$ incident edges, and $d$ edges connecting disjoint pairs of $v$'s neighbors to form triangles.*

It is easy to see that in $H_{r,d}$, each edge has at most one triangle. Hence, there will not be any heavy edges of this kind in the graph. In Kallaugher & Price (2017), the authors show an $\Omega(r\sqrt{d})$ lower bound for the hubs graph.

**Lemma C.4** (Kallaugher & Price (2017))**.** *Given $r$ and $d$, there exist two distributions $\mathcal{G}_1, \mathcal{G}_2$ on subgraphs $G_1$ and $G_2$ of $H_{r,d}$, such that any algorithm which distinguishes them with probability at least $2/3$ requires $\Omega(r\sqrt{d})$ space, where $G_i$ has $\Theta(rd)$ edges and $T(G_1) = d, T(G_2) = 0$.*

Now, given $T$ and $m$, where $T \leq m$, we let $r = \Theta(m/T)$ and $d = T$. We can see $H_{r,d}$ has $\Theta(m)$ edges and $T$ triangles, and we need $\Omega(r\sqrt{d}) = \Omega(m/\sqrt{T})$ space.

When $T > m$, we consider the following $\Omega(m^{3/2}/T)$ lower bound in Bera & Chakrabarti (2017).

Let $H$ be a complete bipartite graph with $b$ vertices on each side (we denote the two sides of vertices by $A$ and $B$). We add an additional $N$ vertex blocks $V_1, V_2, ...V_N$ with each $|V_i| = d$. Alice has an $N$-dimensional binary vector $x$. Bob has an $N$-dimensional binary vector $y$. Both $x$ and $y$ have exactly $N/3$ coordinates that are equal to $1$. Then, we define the edge sets

$$E_{\text{Alice}} = \bigcup_{i:x_i=1} \{\{u,v\}, u \in A, v \in V_i\}$$

and

$$E_{\text{Bob}} = \bigcup_{i:y_i=1} \{\{u,v\}, u \in B, v \in V_i\}\,.$$

Let the final resulting graph be denoted by $G = (V, E)$ where $V = V_H \cup V_1 \cup ... \cup V_N$ and $E = E_H \cup E_{\text{Alice}} \cup E_{\text{Bob}}$.

In Bera & Chakrabarti (2017), the authors show by a reduction to the $\text{DISJ}_N^{N/3}$ problem in communication complexity, that we need $\Omega(N)$ space to distinguish the case when $G$ has 0 triangles from the case when $G$ has at least $b^2 d$ triangles.

Given $m$ and $T$, $T = \Theta(m^{1+\delta})$ where $1 \le \delta < 1/2$, as shown in Bera & Chakrabarti (2017), we can set $b = N^s$ and $d = N^{s-1}$ where $s = 1/(1 - 2\delta)$. This will make $m = \Theta(N^{2s})$ and $T = \Theta(N^{3s-1})$, which will make the $\Omega(m^{3/2}/T)$ lower bound hold. Note that at this time each edge will have an $O(\frac{1}{m})$-fraction of triangles or an $O(\frac{1}{m^{1-\frac{1}{2s}}})$-fraction of triangles. Assume the threshold of the oracle $c = O(m^q)$. When $\delta \ge \max(0, q - \frac{1}{2})$, there will be no heavy edges in the graph.

Theorem C.2 follows from the above discussion.

# D    FOUR-CYCLE COUNTING IN THE ARBITRARY ORDER MODEL

In the 4-cycle counting problem, for each $xy \in E(G)$, define $N_{xy} = |z, w : \{x, y, z, w\} \in \square|$ as the number of 4-cycles attached to edge $xy$.

In this section, we present the one-pass $\widetilde{O}(T^{1/3} + \epsilon^{-2} m/T^{1/3})$ space algorithm behind Theorem 1.3 and the proof of the theorem.[6]

We begin with the following basic idea.

- Initialize: $C \leftarrow 0, S \leftarrow \emptyset$. On the arrival of edge $uv$:
    - With probability $p$, $S \leftarrow \{uv\} \cup S$.
    - $C \leftarrow C + |\{\{w, z\} : uw, wz \text{ and } zv \in S\}|$.
- Return $C/p^3$.

Following the analysis in Vorotnikova (2020), we have

$$\mathbf{Var}[C] \le O(Tp^3 + T\Delta_E p^5 + T\Delta_W p^4) \,,$$

where $\Delta_E$ and $\Delta_W$ denote the maximum number of 4-cycles sharing a single edge or a single wedge.

There are two important insights from the analysis: first, this simple sampling scheme can output a good estimate to the number of 4-cycles on graphs that do not have too many heavy edges and heavy wedges (the definitions of heavy will be shown later). Second, assuming that we can store all the heavy edges, then at the end of the stream we can also estimate the number of 4-cycles that have exactly one heavy edge but do not have heavy wedges.

For the 4-cycles that have heavy wedges, we use an idea proposed in McGregor & Vorotnikova (2020); Vorotnikova (2020): for any node pair $u$ and $v$, if we know their number of common neighbors (denoted by $k$), we can compute the exact number of 4-cycles with $u, v$ as a diagonal pair (i.e., the four cycle contains two wedges with the endpoints $u, v$), and this counts to $\binom{k}{2}$. Furthermore, if $k$ is large (in this case we say all the wedges with endpoints $u, v$ are heavy), we can detect it and obtain a good estimation to $k$ by a similar vertex sampling method mentioned in Section 3. We can then estimate the total number of 4-cycles that have heavy wedges with our samples.

At this point, we have not yet estimated the number of 4-cycles having more than one heavy edge but without heavy wedges. However, McGregor & Vorotnikova (2020) shows this class of 4-cycles only takes up a small fraction of $T$, and hence we can get a $(1 \pm \epsilon)$-approximation to $T$ even without counting this class of 4-cycles.

The reader is referred to Algorithm 5 for a detailed version of our one-pass, 4-cycle counting algorithm. Before the stream, we randomly select a node set $S$, where each vertex is in $S$ with probability

---

[6]We assume $\frac{1}{T^{1/6}} \le O(\epsilon)$, which is the same assumption as in previous work

---

**Algorithm 5** Counting 4-cycles in the Arbitrary Order Model

---

1: **Input:** Arbitrary Order Stream and an oracle that outputs TRUE if $N_{xy} \geq T/\rho$ and FALSE otherwise.
2: **Output:** An estimate of the number $T$ of 4-cycles.
3: Initialize $A_l \leftarrow 0$, $A_h \leftarrow 0$, $A_w \leftarrow 0$ $E_L \leftarrow \emptyset$ $E_H \leftarrow \emptyset$, $E_S \leftarrow \emptyset$, and $W \leftarrow \emptyset$. Set $\rho \leftarrow T^{1/3}$, $p \leftarrow \alpha\epsilon^{-2}\log n/\rho$ for a sufficiently large $\alpha$, and Let $S$ be a random subset of nodes such that each node is in $S$ with probability $p$.
4: **while** seeing edge $uv$ in the stream **do**
5:     **if** $u \in S$ or $v \in S$ **then**
6:         $E_S \leftarrow \{uv\} \cup E_S$ .
7:     **if** the oracle outputs FALSE **then**
8:         With probability $p$, $E_L \leftarrow \{uv\} \cup E_L$ .
9:         Find pair $(w, z)$ such that $uw, wz$, and $zv \in E_L$, let $D \leftarrow D \cup \{(u, w, z, v)\}$ .
10:     **else**
11:         $E_H \leftarrow \{uv\} \cup E_H$ .
12: **for** each node pair $(u, v)$ **do**
13:     let $q(u, v)$ be the number of wedges with center in $S$ and endpoints $u$ and $v$.
14:     **if** $q(u, v) \geq pT^{1/3}$ **then**
15:         $A_w \leftarrow A_w + \binom{q(u,v)}{2}$ .
16:         $W \leftarrow W \cup \{(u, v)\}$ .
17: **for** each 4-cycle $d$ in $D$ **do**
18:     **if** the end points of wedges in $d$ are not in $W$ **then**
19:         $A_l \leftarrow A_l + 1$
20: **for** each edge $uv$ in $E_H$ **do**
21:     **for** each 4-cycle $d$ formed with $uv$ and $e \in E_L$ **do**
22:         **if** the end points of wedges in $d$ are not in $W$ **then**
23:             $A_h \leftarrow A_h + 1$
24: **return:** $A_l/p^3 + A_h/p^3 + A_w$ .

---

$p$ independently, and we later store all edges that are incident to $S$ during the stream. In the meantime, we define the edge $uv$ to be *heavy* if $N_{uv} \geq T^{2/3}$ and train the oracle to predict whether $uv$ is heavy or not when we see the edge $uv$ during the stream. Let $p = \widetilde{O}(\epsilon^{-2}/T^{1/3})$. We store all the heavy edges and sample each light edge with probability $p$ during the stream. Upon seeing see a light edge $uv$, we look for the 4-cycles that are formed by $uv$ and the light edges that have been sampled before, and then record them in set $D$. At the end of the stream, we first find all the node pairs that share many common neighbors in $S$ and identify them as heavy wedges. Then, for each 4-cycle $d \in D$, we check if $d$ has heavy wedges, and if so, remove it from $D$. Finally, for each heavy edge $uv$ indicated by the oracle, we compute the number of 4-cycles that are formed by $uv$ and the sampled light edges, and without heavy wedges. This completes all parts of our estimation procedure.

We now prove Theorem 1.3, restated here for the sake of convenience.

**Theorem 1.3.** *There exists a one-pass algorithm, Algorithm 5, with space complexity $\widetilde{O}(T^{1/3} + \epsilon^{-2}m/T^{1/3})$ in the arbitrary order model that, using a learning-based oracle, returns a $(1 \pm \epsilon)$-approximation to the number $T$ of four cycles with probability at least $7/10$.*

*Proof.* From the Lemma 3 in Vorotnikova (2020), we know that with probability at least $9/10$, we can get an estimate $A_w$ such that
$$A_w = T_w \pm \epsilon T.$$

Where $T_w$ is the number of the 4-cycles that have at least one heavy wedge. We note that if a 4-cycle has two heavy wedges, it will be counted twice. However, Vorotnikova (2020) shows that this double counting is at most $O(T^{2/3}) = O(\epsilon)T$.

For the edge sampling algorithm mentioned in D, from the analysis in Vorotnikova (2020), we have
$$\mathbf{Var}[A_l/p^3] \leq O(T/p^3 + T\Delta_E/p + T\Delta_W/p^2) .$$

Notice that in our algorithm, the threshold of the heavy edges and heavy wedges are $N_{uv} \geq T^{2/3}$ and $N_w \geq T^{1/3}$, respectively, which means $\mathbf{Var}[\frac{A_l}{p^3}] = O(\epsilon^2 T^2)$. Hence, we can get an estimate $A_l$ such that with probability at least $9/10$,

$$A_l = T_l \pm \epsilon T \,,$$

where $T_l$ is the number of 4-cycles that have no heavy edges. Similarly, we have with probability at least $9/10$,

$$A_h = T_h \pm \epsilon T \,,$$

where $T_h$ is the number of 4-cycles that have at most one heavy edge.

One issue is that the algorithm has not yet estimated the number of 4-cycles having more than one heavy edge, but without heavy wedges. However, from Lemma 5.1 in McGregor & Vorotnikova (2020) we get that the number of this kind of 4-cycles is at most $O(T^{5/6}) = O(\epsilon)T$. Hence putting everything together, we have

$$|A_l/p^3 + A_h/p^3 + A_w - T| \leq O(\epsilon)T \,,$$

which is what we need.

Now we analyze the space complexity. The expected number of light edges we sample is $O(mp) = \widetilde{O}(\epsilon^{-2}m/T^{1/3})$ and the expected number of nodes in $S$ and edges in $E_S$ is $O(2mp) = \widetilde{O}(\epsilon^{-2}m/T^{1/3})$. The expected number of 4-cycles we store in $D$ is $O(Tp^3) = \widetilde{O}(\epsilon^{-6})$. We store all the heavy edges, and the number of heavy edges is at most $O(T^{1/3})$. Therefore, the expected total space is $\widetilde{O}(T^{1/3} + \epsilon^{-2}m/T^{1/3})$. $\qquad \square$

## E   RUNTIME ANALYSIS

In this section, we verify the runtimes of our Algorithms 1, 2, 3, 4, and 5.

**Proposition E.1.** *Algorithm 1 runs in expected time at most $\widetilde{O}\left(\min(\epsilon^{-2}m^{5/3}/T^{1/3}, \epsilon^{-1}m^{3/2})\right)$ in the setting of Theorem 1.1, or $\widetilde{O}\left(\min(\epsilon^{-2}m^{5/3}/T^{1/3}, K \cdot \epsilon^{-1}m^{3/2})\right)$ in the setting of Theorem 1.4.*

*Proof.* For each edge $ab \in S_L \cup S_M$, we check whether we see both $va$ and $vb$ (lines 9-10) when looking at edges adjacent to $v$. So, this takes time $|S_L| + |S_M|$ per edge in the stream. We similarly check for each edge in $S_{aux}$ (lines 11-12), so this takes time $|S_{aux}|$. Finally, for each edge $vu$ (line 13), we note that lines 14-17 can trivially be implemented in $O(1)$ time, along with 1 call to the oracle. The remainder of the oracle simply involves looking through the set $S_{aux}$ and $S_M$ to search or count for elements. Thus, the overall runtime is $O(|S_L| + |S_M| + |S_{aux}|)$ per stream element, so the runtime is $O\left(m \cdot (|S_L| + |S_M| + |S_{aux}|)\right)$.

This is at most $O(m \cdot S)$, where $S$ is the space used by the algorithm. So, the runtime is $\widetilde{O}\left(\min(\epsilon^{-2}m^{5/3}/T^{1/3}, \epsilon^{-1}m^{3/2})\right)$ in the setting of Theorem 1.1, and is at most $\widetilde{O}\left(\min(\epsilon^{-2}m^{5/3}/T^{1/3}, K \cdot \epsilon^{-1}m^{3/2})\right)$ in the setting of Theorem 1.4. $\qquad \square$

**Proposition E.2.** *Algorithm 2 runs in time $\widetilde{O}(\epsilon^{-2}(\alpha + m\beta/T)m)$.*

*Proof.* For each edge $ab \in S^i$ for each $0 \leq i \leq c\epsilon^{-2}$, we check whether we see both $ya$ and $yb$ in the stream when looking at edges adjacent to $y$. So, lines 7-8 take time $O(\sum |S^i|)$ for each edge we see in the stream. By storing each $S^i$ (and each $Q^i$) in a balanced binary search tree or a similar data structure, we can search for $xy \in S^i$ in time $O(\log n)$ in line 11, and it is clear that lines 12-16 take time $O(\log n)$ (since insertion into the data structure for $Q^i, S^i$ can take time $O(\log n)$). Since we do this for each $i$ from 0 to $c\epsilon^{-2}$ and for each incident edge $yx$ we see, in total we spend $O(\epsilon^{-2} \log n + \sum |S^i|)$ time up to line 16. The remainder of the lines take time $O(\epsilon^{-2} \cdot m \log n)$, since the slowest operation is potentially deleting an edge from each $S^i$ and a value from each $Q^i$ up to $m$ times (for each edge).

Overall, the runtime is $\widetilde{O}(\epsilon^{-2} \cdot m + m \cdot \sum |S^i|) = \widetilde{O}(m \cdot S)$, where $S$ is the total space used by the algorithm. Hence, the runtime is $\widetilde{O}(\epsilon^{-2}(\alpha + m\beta/T)m)$. $\qquad \square$

**Proposition E.3.** *Algorithm 3 runs in time* $\widetilde{O}(K\epsilon^{-2}(\alpha + m\beta/T)m)$.

*Proof.* For each edge $ab \in S_i^j$ for each $1 \le i \le O(\log n), 1 \le j \le O(\log\log n)$, we check whether we see both $ya$ and $yb$ in the stream when looking at edges adjacent to $y$. So, line 5 takes time $O(\sum |S_i^j|)$ for each edge we see in the stream. By storing each $S_i^j$ in a balanced binary search tree or a similar data structure, we can search for $xy \in S^i$ in time $O(\log n)$ in line 7, and it is clear that lines 8-12 take time $O(\log n)$ (since insertion into the data structure for $Q_i^j$ can take time $O(\log n)$). Since we do this for each $i$ from 0 to $c\epsilon^{-2}$ and for each incident edge $yx$ we see, in total we spend poly $\log n \cdot O(\sum |S^i|)$ time up to line 12. Finally, lines 13-15 take time poly $\log n$.

Overall, the runtime is $\widetilde{O}(m \cdot \sum |S^i|) = \widetilde{O}(m \cdot S)$, where $S$ is the total space used by the algorithm. Hence, the runtime is $\widetilde{O}(K\epsilon^{-2}(\alpha + m\beta/T)m)$. $\qquad\square$

**Proposition E.4.** *Algorithm 4 runs in time* $\widetilde{O}\left(\epsilon^{-1} \cdot m^2/\sqrt{T} + \epsilon^{-1} \cdot m^{3/2}\right)$.

*Proof.* For each edge $e$ in the stream, first we check the oracle, and we either add $e$ to $H$ or to $S_L$ with probability $p$ (lines 6-9), which take $O(1)$ time. The remaining lines (lines 10-18) involve operations that take $O(1)$ time for each $w$ which represents a pair $(e_1, e_2)$ of edges where $e_1, e_2 \in S_L \cup H$ and $(e, e_1, e_2)$ forms a triangle. So, the runtime is bounded by the amount of time it takes to find all $e_1, e_2 \in S_L \cup H$ such that $(e, e_1, e_2)$ forms a triangle. If the edge $e = (u, v)$, we just find all edges in $S_L \cup H$ adjacent to $u$, and all edges in $S_L \cup H$ adjacent to $v$. Then, we sort these edges based on their other endpoint and match the edges if they form triangles. So, the runtime is $\widetilde{O}(|S_L| + |H|)$ per edge $e$ in the stream. Finally, line 19 takes O(1) time.

Overall, the runtime is $\widetilde{O}(m \cdot (|S_L| + |H|)) = \widetilde{O}(m \cdot S)$ where $S$ is the total space of the algorithm. So, the runtime is $\widetilde{O}\left(\epsilon^{-1} \cdot m^2/\sqrt{T} + \epsilon^{-1} \cdot m^{3/2}\right)$. $\qquad\square$

**Proposition E.5.** *Algorithm 5 runs in time* $\widetilde{O}\left(\epsilon^{-2}/T^{1/3} \cdot (n^3 + m^2)\right)$.

*Proof.* We note that for each edge $uv$ in the stream (line 4), the code in the loop (lines 5-11) can be implemented using 1 oracle call and $\widetilde{O}(|E_L|)$ time. The only nontrivial step here is to find pairs $(w, z)$ such that $uw, wz, zv$ are all in $E_L$. However, by storing $E_L$ in a balanced binary search tree or similar data structure, one can enumerate through each edge $wz \in E_L$ and determine if $uw$ and $zv$ are in $E_L$ in $O(\log |E_L|)$ time. So, lines 4-11 take time $\widetilde{O}(|E_L|)$ per stream element.

Next, lines 12-16 can easily be implemented in time $O(n^2 \cdot |S|)$, lines 17-19 can be easily implemented in time $\widetilde{O}(|D|)$ if the vertices in $W$ are stored properly, and lines 20-23 can be done in $\widetilde{O}(|E_H| \cdot |E_L|)$ time. The last part is true since we check each $uv \in E_H$ and $e = (u', v') \in E_L$, and then check if $u, u', v', v$ form a 4-cycle by determining if $u, u'$ and $v, v'$ are in $E_L \cup E_H$ (which takes time $O(\log |E_L| + \log |E_H|)$ time assuming $E_L, E_H$ are stored in search tree or similar data structure).

Overall, we can bound the runtime as $\widetilde{O}(m \cdot |E_L| + n^2 \cdot |S| + |D| + |E_H| \cdot |E_L|) = \widetilde{O}(m \cdot |E_L| + n^2 \cdot |S|)$. As each edge is in $E_L$ with probability at most $p$ and each edge is in $S$ with probability at most $p$, the total runtime is $\widetilde{O}\left((m^2 + n^3) \cdot p\right) = \widetilde{O}\left(\epsilon^{-2}/T^{1/3} \cdot (n^3 + m^2)\right)$. $\qquad\square$

# F  ADDITIONAL EXPERIMENTAL RESULTS

All of our graph experiments were done on a CPU with i5 2.7 GHz dual core and 8 GB RAM or a CPU with i7 1.9 GHz 8 core and 16GB RAM. The link prediction training was done on a single GPU.

## F.1  DATASET DESCRIPTIONS

- **Oregon:** 9 graphs sampled over 3 months representing a communication network of internet routers Leskovec & Krevl (2014); Leskovec et al. (2005).

- **CAIDA:** 98 graphs sampled approximately weekly over 2 years, representing a communication network of internet routers Leskovec & Krevl (2014); Leskovec et al. (2005).
- **Reddit Hyperlinks:** Network where nodes represent sub communities (subreddits) and edges represent posts that link two different sub communities Leskovec & Krevl (2014); Kumar et al. (2018).
- **WikiBooks:** Network representing Wikipedia users and pages, with editing relationships between them Rossi & Ahmed (2015).
- **Twitch:** A user-user social network of gamers who stream in a certain language. Vertices are the users themselves and the links are mutual friendships between them. Rozemberczki et al. (2019)
- **Wikipedia:** This is a co-occurrence network of words appearing in the first million bytes of the Wikipedia dump. Mahoney (2011)
- **Synthetic Power law:** Power law graph sampled from the Chung-Lu-Vu (CLV) model with the expected degree of the $i$th vertex proportional to $1/i^2$ Chung et al. (2003). To create this graph, the vertices are 'revealed' in order. When the $j$th vertex arrives, the probability of forming an edge between the $j$th vertex and $i$th vertex for $i < j$ is proportional to $1/(ij)^2$.

## F.2 DETAILS ON LINK PREDICTION ORACLE

Our oracle for the Twitch and Wikipedia graphs is based on Link Prediction, where the task is to estimate the likelihood of the existence of edges or to find missing links in the network. Here we use the method and code proposed in Zhang & Chen (2018). For each target link, it will extract a local enclosing subgraph around a node pair, and use a Graph Neural Network to learn general graph structure features for link prediction. For the Twitch network, we use all the training links as the training data for the graph neural network, while for the Wikipedia network, we use $30\%$ of the training links as the training data due to memory limitations. For the Twitch network, our set of links that we will try to predict are between two nodes that have a common neighbor in the training network, but do not form a link in the training network. This is about 3.8 million pairs, and we call this the candidate set. We do this for memory considerations. For the remaining node pairs, we set the probability they form a link to be 0. For the Wikipedia network, we randomly select a link candidate set of size 20 times the number of edges (about 1 million pairs) from the entire set of testing links. These link candidate sets will be used by the oracle to determine heaviness. Then, we use this network to do our prediction for all links in our candidate sets for the two networks.

Now we are ready to build our heavy edge oracle. For the adjacency list model, when we see the edge $uv$ in the stream, we know all the neighbors of $u$, and hence, we only need to predict the neighbors of $v$. Let $\deg(v)$ be the degree of $v$ in the training graph. The training graph and testing graph are randomly split. Hence, we can use the training set to provide a good estimation to the degree of $v$. Next, we choose the largest $\deg(v)$ edges incident to $v$ from the candidate set, in terms of their predicted likelihood given by the the neural network, as our prediction of $N(v)$, which leads to an estimate of $R_{uv}$. For the arbitrary order model, we use the same technique as above to predict $N(u)$ and $N(v)$, and estimate $N_{uv}$.

## F.3 PARAMETER SELECTION FOR ALGORITHM 2

We need to set two parameters for Algorithm 2: $p$, which is the edge sampling probability, and $\theta$, which is the heaviness threshold. In our theoretical analysis, we assume knowledge of a lower bound on $T$ in order to set $p$ and $\theta$, as is standard in the theoretical streaming literature. However, in practice, such an estimate may not be available; in most cases, the only parameter we are given is a space bound for the number of edges that can be stored. To remedy this discrepancy, we modify our algorithm in experiments as follows.

First, we assume we only have access to the stream, a space parameter $Z$ indicating the maximum number of edges that we are allowed to store, and an estimate of $m$, the number of edges in the stream. Given $Z$, we need to designate a portion of it for storing heavy edges, and the rest for storing light edges. The trade-off is that the more heavy edges we store, the smaller our sampling probability of light edges would be. We manage this trade off in our implementation by reserving $0.3 \cdot Z$ of the edge 'slots' for heavy edges. The constant 0.3 is *fixed* throughout all our experiments.

We then set $p = 0.7 \cdot Z/m$. To improve the performance of our algorithm, we optimize for space usage by *always* insuring that we are storing exactly $Z$ space (after observing the first $Z$ edges of the stream). To do so and still maintain our theoretical guarantees, we perform the following procedure. Call the first $Z$ edges of the stream the early phase and the rest of the edges the late phase.

We always keep edges in the early phase to use our space allocation $Z$ and also keep track of the 0.3-fraction of the heaviest edges. After the early phase is over, i.e., more than $Z$ edges have passed in the stream, if a new incoming edge is heavier than the lightest of the stored heavy edges, we replace the least heavy stored edge with the new arriving edge and re-sample the replaced edge with probability $p$. Otherwise, if the new edge is not heavier than the lightest stored edge, we sample the new incoming edge with probability $p$. If we exceed $Z$, the space threshold, we replace one of the light edges sampled in the early phase. Then similarly as before, we re-sample this replaced edge with probability $p$ and continue this procedure until one of the early light edges has been evicted. In the case that there are no longer any of the light edges sampled in the early phase stored, we replace a late light edge and then any arbitrary edge (again performing the re-sampling procedure for the evicted edge).

Note that in our modification, we only require our predictor be able to compare the heaviness between two edges, i.e., we do not need the predictor to output an estimate of the number of triangles on an edge. This potentially allows for more flexibility in the choice of predictors.

If the given space bound meets the space requirements of Theorem C.1, then the theoretical guarantees of this modification simply carry over from Theorem 1.2: we always keep the heaviest edges and always sample light edges with probability at least $p$. In case the space requirements are not met, the algorithm stores the most heavy edges as to reduce the overall variance.

### F.4 ADDITIONAL FIGURES FROM ARBITRARY ORDER TRIANGLE COUNTING EXPERIMENTS

Additional figures for our arbitrary order triangle counting experiments are given in Figures 4 and 5.

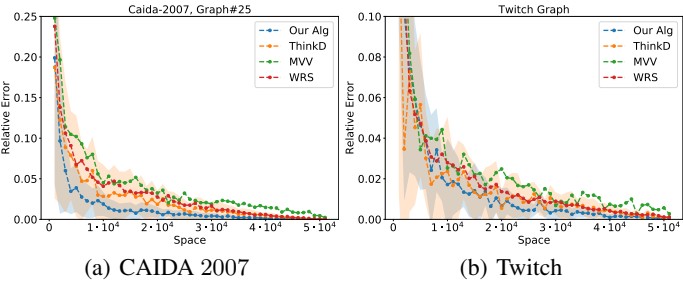

(a) CAIDA 2007          (b) Twitch

Figure 4: Error as a function of space for various graph datasets.

### F.5 EXPERIMENTAL DESIGN FOR ADJACENCY LIST EXPERIMENTS

We now present our adjacencly list experiments. At a high level overview, similarly to the arbitrary order experiments, for our learning-based algorithm, we reserve the top 10% of the total space for storing the heavy edges. To do this in practice, we can maintain the heaviest edges currently seen so far and evict the smallest edge in the set when a new heavy edge is predicted by the oracle and we no longer have sufficient space. We also consider a multi-layer sub-sampling version of the algorithm in Section B.2. Here we use more information from the oracle by adapting the sub-sampling rates of edges based on their predicted value. For more details, see Section F.5. Our results are presented in Figure 6 (with additional plots given in Figure 7). Our algorithms soundly outperform the MVV baseline for most graph datasets. We only show the error bars for the multi-layer algorithm and MVV for clarity. Additional details follow.

We use the same predictor for $N_{xy}$ in Section 4 as a prediction for $R_{xy}$. The experiment is done under a random vertex arrival order. For the learning-based algorithm, suppose $Z$ is the maximum number of edges that we are allowed to store. we set the $k = Z/10$ edges with the highest predicted $R_{uv}$ values to be the heavy edges, and store them during the stream (i.e., we use 10% of the total

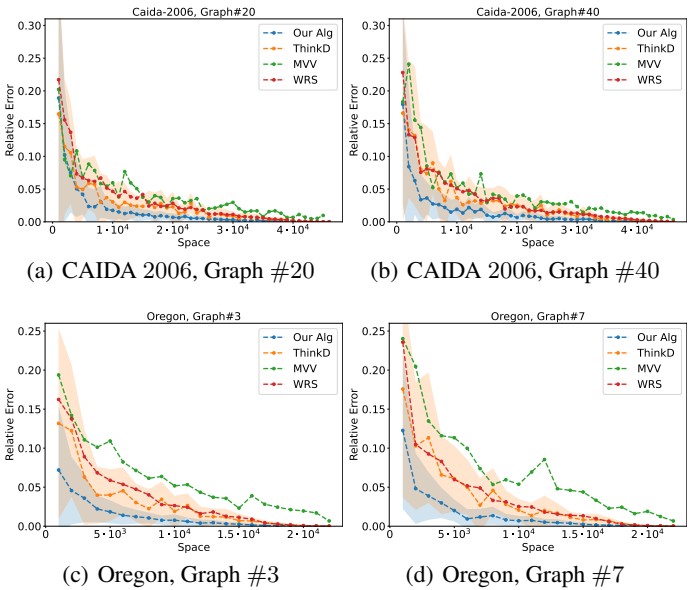

Figure 5: Error as a function of space for various graph datasets.

space for storing the heavy edges). For the remaining edges, we use a similar sampling-based method. Note that it is impossible to know the $k$-heaviest edges before we see all the edges in the stream. However, in the implementation we can maintain the $k$ heaviest edges we currently have seen so far, and evict the smallest edge in the set when a new heavy edge is predicted by the oracle.

We also consider the multi-layer sub-sampling algorithm mentioned in Section B.2, which uses more information from the oracle. We notice that in many graph datasets, most of the edges having very few number of triangles attached to. Taking an example of the Oregon and CAIDA graph, only about 3%-5% of edges will satisfy $R_e \geq 5$ under a random vertex arrival order. Hence, for this edges, intuitively we can estimate them using a slightly smaller space.

For the implementation of this algorithm(we call it multi-layer version), we use $10\%$ of the total space for storing the top $k = Z/10$ edges, and $70\%$ of the space for sub-sampling the edges that the oracle predict value is very tiny(the threshold may be slightly different for different datasets, like for the Oregon and CAIDA graph, we set the threshold to be 5). Then, we use $20\%$ of the space for sub-sampling the remaining edges, for which we call them the medium edges.

## F.6 Figures from Adjacency List Triangle Counting Experiments

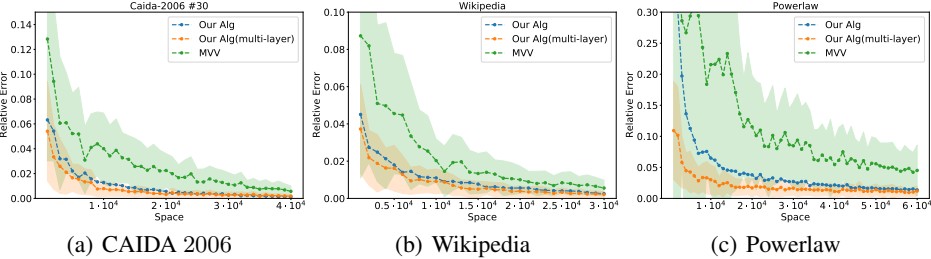

Figure 6: Error as a function of space in the adjacency list model.

Additional figures from the adjacency list triangle counting experiments are shown in Figure 7. They are qualitatively similar to the results presented in Figure 6 as the multi-layer sampling algorithm is superior over the MVV baseline for all of our datasets.

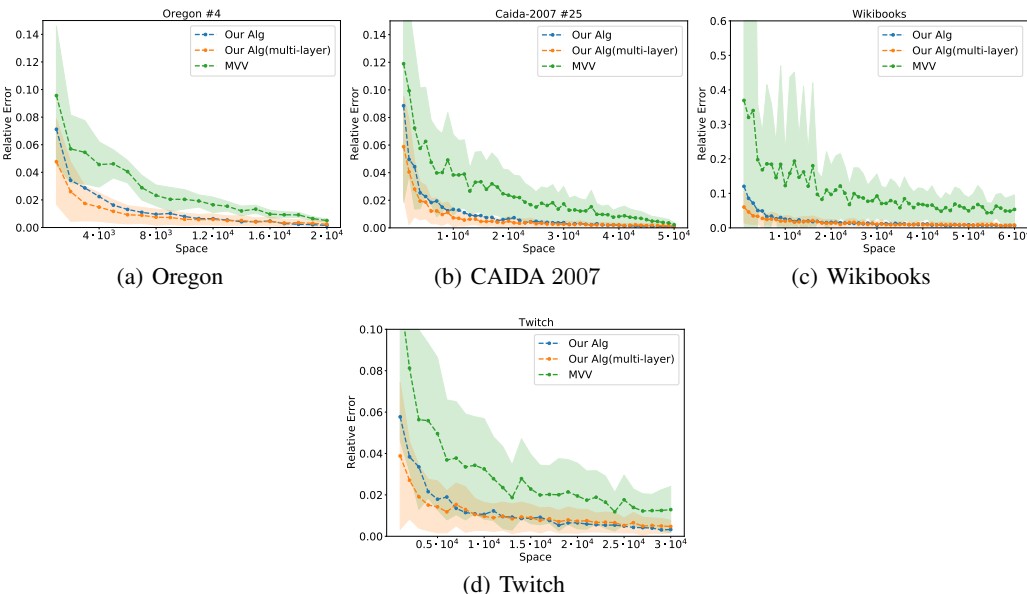

Figure 7: Error as a function of space for various graph datasets.

## F.7 Accuracy of the Oracle

In this section, we evaluate the accuracy of the predictions the oracle gives in our experiments.

**Value Prediction Oracle** : We use the prediction of $N_{xy}$ in Section 4 as a value prediction for $R_{xy}(x \leq_s y)$, under a fixed random vertex arrival order. The results are shown in Figure 8. For a fixed approximation factor $k$, we compute the failure probability $\delta$ of the value prediction oracle as follows: $\delta = \#/m$, where $m$ is the number of total edges and $\#$ equals to the number of edges $e$ that $p(e) \geq k\alpha R_e + \beta$ or $p(e) \leq \frac{1}{k}R_e - \beta$, respectively. Here we set $\alpha = 1, \beta = 10$ for all graph datasets.

We can see for the smaller side, there are very few edges $e$ such that $p(e) \leq \frac{1}{k}R_e - \beta$. This meets the assumption of the exponential decay tail bound of the error. For the larger side, it also meets the assumption of the linear decay tail bound on most of the graph datasets.

## F.8 Details on Oracle Training Overhead

The overhead of the oracles used in our experiments vary from task to task. For the important use case illustrated by the Oregon and CAIDA datasets in which we are interested in repeatedly counting triangles over many related streams, we can pay a small upfront cost to create the oracle which can be reused over and over again. Thus, the time complexity of building the oracle can be amortized over many problem instances, and the space complexity of the oracle is relatively small as we only need to store the top $10\%$ of heavy edges from the first graph (a similar strategy is used in prior work on learning-augmented algorithms in Hsu et al. (2019b)). To give more details, for this snapshot oracle, we simply calculate the $N_e$ values for all edges only in the first graph. We then keep the heaviest edges to form our oracle. Note that this takes polynomial time to train. The time complexity of using this oracle in the stream is as follows: when an edge in the stream comes, we simply check if it's among the predicted heavy edges in our oracle. This is a simple lookup which can even be done in constant time using hashing.

For learning-based oracles like the linear regression predictor, we similarly need to pay an upfront cost to train the model, but the space required to store the trained model depends on the dimension of the edge features. For the Reddit dataset with $\sim 300$ features, this means that the storage cost for the oracle is a small fraction of the space of the streaming algorithm. Training of this oracle can be done in polynomial time and can even be computed in a stream via sketching and sampling

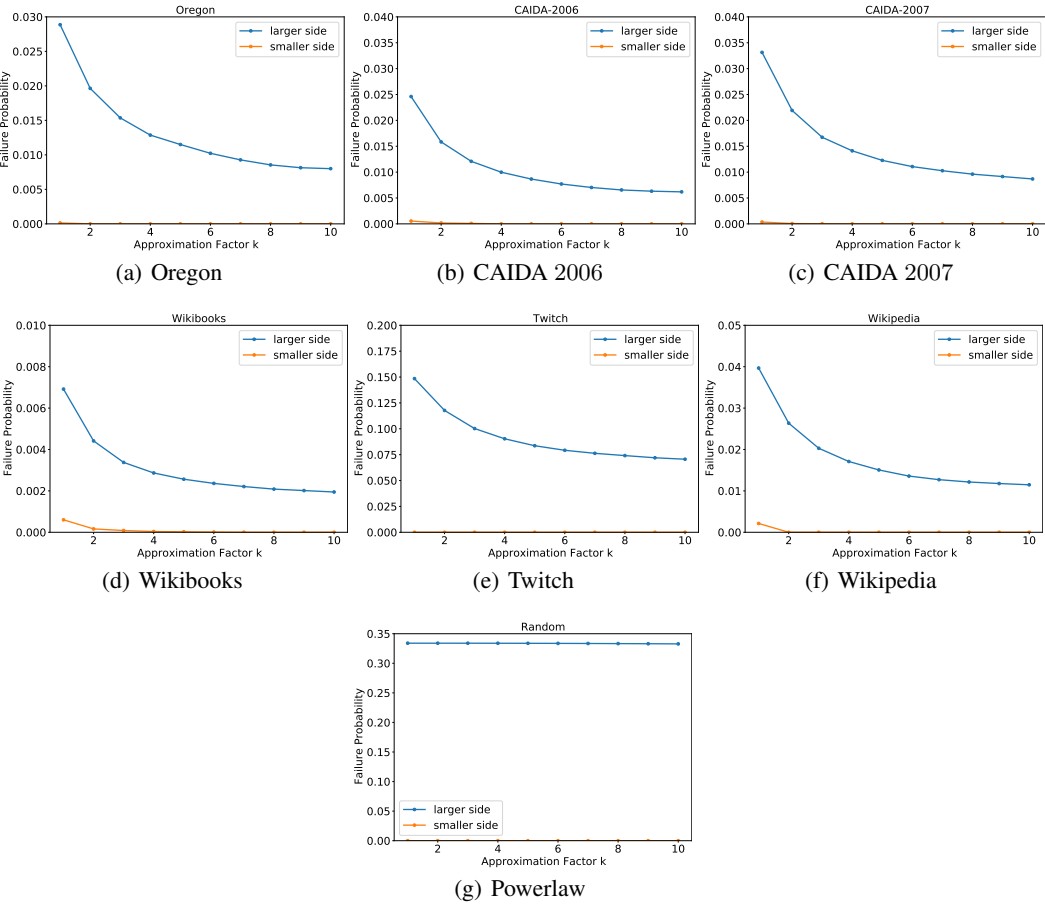

Figure 8: Failure probability as a function of approximation factor $k$ for various graph datasets.

techniques which reduce the number of constraints from m (number of edges) to roughly linear in the number of features.

Our expected value oracle exploits the fact that our input graph is sampled from the CLV random graph model. Given this, we can explicitly calculate the expected value of $N_e$ for every edge which requires no training time and nearly constant space. For training details for our link prediction model, see Section F.2. Note that in general, there is a wide and rich family of predictors which can be trained using sublinear space or in a stream such as regression Woodruff (2014), classification for example using SVMs Andoni et al. (2020); Rai et al. (2009) and even deep learning modelsGomes et al. (2019).

## G  IMPLICIT PREDICTOR IN PRIOR WORKS

We prove that the first pass of the two pass triangle counting Algorithm given in Section 3.1 of McGregor et al. (2016), satisfies the conditions of the $K$-noisy oracle in Definition 1.1. Therefore, our work can be seen as a generalization of their approach when handling multiple related data sets, where instead of performing two passes on each data set and using the first of which to train the heavy edge oracle, we perform the training once according to the first related dataset, and we get a one pass algorithm for all remaining sets.

We first recall the first pass of (McGregor et al., 2016, Section 3.1):

1. Sample each node $z$ of the graph with probability $p = C\epsilon^{-2} \log m/\rho$ for some large constant $C > 0$. Let $Z$ be the set of sampled nodes.

2. Collect all edges incident on the set $Z$.

3. For any edge $e = \{u, v\}$, let $\widetilde{t}(e) = |\{z \in Z : u, v \in N(z)\}|$ and define the oracle as

$$\text{oracle}(e) = \begin{cases} \text{LIGHT} & \text{if } \widetilde{t}(e) < p \cdot \rho \\ \text{HEAVY} & \text{if } \widetilde{t}(e) \geq p \cdot \rho. \end{cases}$$

**Lemma G.1.** *For any edge* $e = (u, v)$*,* $\text{oracle}(e) = \text{LIGHT}$ *implies* $N_e \leq 2\rho$ *and* $\text{oracle}(e) = \text{HEAVY}$ *implies* $N_e > \rho/\sqrt{2}$ *with failure probability at most* $1/n^{10}$*.*

*Proof.* For any edge $e$, it follows that $\widetilde{t}(e) \sim \text{Bin}(N_e, p)$. Therefore if $N_e > 2\rho$,

$$\Pr[\widetilde{t}(e) < p \cdot \rho] \leq \exp(-\Omega(p \cdot \rho)) \leq 1/n^{10}$$

by picking $C$ large enough in the definition of $p$. The other case follows similarly. $\qquad\square$

**Lemma G.2.** *The expected space used by the above oracle is* $O(pm)$*.*

*Proof.* Each vertex is sampled in $Z$ with probability $p$ and we keep all of its incident edges. Therefore, the expected number of edges saved is $O(p \sum_v d_v) = O(pm)$. $\qquad\square$

Note that the oracle satisfies the conditions of the $K$-noisy oracle in Definition 1.1. For example, if $N_e \geq C'\rho$ for $C' \gg 1$, Definition 1.1 only assumes that we incorrectly classify $e$ with probability $1/C'$, whereas the oracle presented above incorrectly classifies $e$ with probability $\exp(-C')/n^{10}$ which is much smaller than the required $1/C'$.

# H   LEARNABILITY RESULTS

In this section, we give formal learning bounds for efficient predictor learning for edge heaviness. In particular, we wish to say that a good oracle or predictor for edge heaviness and related graph parameters can be learned efficiently using few samples if we observe graph instances drawn from a distribution. We can view the result of this section, which will be derived via the PAC learning framework, as one formalization of data driven algorithm design. Our results are quite general but we state simple examples throughout the exposition for clarity. Our setting is formally the following.

Suppose there is an underlying distribution $\mathcal{D}$ which generates graph instances $H_1, H_2, \cdots$ all on $n$ vertices. Note that this mirrors some of our experimental setting, in particular our graph datasets which are similar snapshots of a dynamic graph across time.

Our goal is to efficiently learn a good predictor $f$ among some family of functions $\mathcal{F}$. The input of each $f$ is a graph instance $H$ and the output is a feature vector in $k$ dimensions. The feature vector represents the prediction of the oracle and can encapsulate a variety of different meanings. One example is when $k = |E|$ and $f$ outputs an estimate of edge heaviness for all edges. Another is when $k << |E|$ and $f$ outputs the id's of the $k$ heaviest edges. We also think of each input instance $H$ as encoded in a vector in $\mathbb{R}^p$ for $p \geq \binom{n}{2}$ (for example, each instance is represented as an adjacency matrix). Note that we allow for $p > \binom{n}{2}$ if for example, each edge or vertex for $H \sim \mathcal{D}$ also has an endowed feature vector.

To select the 'best' $f$, we need to precisely define the meaning of best. Note that in many settings, this involves a loss function which captures the quality of a solution. Indeed, suppose we have a loss function $L : f \times H \to \mathbb{R}$ which represents how well a predictor $f$ performs on some input $H$. An example of $L$ could be squared error from the output of $f$ to the true edge heaviness values of edges in $H$. Note that such a loss function clearly optimizes for predicting the heavy edges well.

Our goal is to learn the best function $f \in \mathcal{F}$ which minimizes the following objective:

$$\mathbb{E}_{H \sim D}[L(f, H)]. \tag{1}$$

Let $f^*$ be such the optimal $f \in \mathcal{F}$, and assume that for each instance $H$ and each $f \in F$, $f(H)$ can be computed in time $T(p, k)$. For example, suppose graphs drawn from $\mathcal{D}$ possess edge features in $\mathbb{R}^d$, and that our family $\mathcal{F}$ is parameterized by a single vector $\theta \in \mathbb{R}^d$ and represents linear functions

which outputs the dot product of each edge feature with $\theta$. Then it is clear that $T(p, k)$ is a (small) polynomial in the relevant parameters.

Our main result is the following.

**Theorem H.1.** *There is an algorithm which after $poly(T(p,k), 1/\epsilon)$ samples, returns a function $\hat{f}$ that satisfies*

$$\mathbb{E}_{H \sim D}[L(\hat{f}, H)] \leq \mathbb{E}_{H \sim D}[L(f^*, H)] + \epsilon$$

*with probability at least $9/10$.*

We remark that one can boost the probability of success to $1 - \delta$ by taking additional $\log(1/\delta)$ multiplicative samples.

The above theorem is a PAC-style bound which shows that only a small number of samples are needed in order to ensure a good probability of learning an approximately-optimal function $\hat{f}$. The algorithm to compute $\hat{f}$ is the following: we simply minimize the empirical loss after an appropriate number of samples are drawn, i.e., we perform empirical risk minimization. This result is proven by Theorem H.3. Before introducing it, we need to define the concept of pseudo-dimension for a function class which is the more familiar VC dimension, generalized to real functions.

**Definition H.2** (Pseudo-Dimension, Definition 9 Lucic et al. (2018)). *Let $\mathcal{X}$ be a ground set and $\mathcal{F}$ be a set of functions from $\mathcal{X}$ to the interval $[0, 1]$. Fix a set $S = \{x_1, \cdots, x_n\} \subset \mathcal{X}$, a set of real numbers $R = \{r_1, \cdots, r_n\}$ with $r_i \in [0, 1]$ and a function $f \in \mathcal{F}$. The set $S_f = \{x_i \in S \mid f(x_i) \geq r_i\}$ is called the induced subset of $S$ formed by $f$ and $R$. The set $S$ with associated values $R$ is shattered by $\mathcal{F}$ if $|\{S_f \mid f \in \mathcal{F}\}| = 2^n$. The pseudo-dimension of $\mathcal{F}$ is the cardinality of the largest shattered subset of $\mathcal{X}$ (or $\infty$).*

The following theorem relates the performance of empirical risk minimization and the number of samples needed, to the notion of pseudo-dimension. We specialize the theorem statement to our situation at hand. For notational simplicity, we define $\mathcal{A}$ be the class of functions in $f$ composed with $L$:

$$\mathcal{A} := \{L \circ f : f \in \mathcal{F}\}.$$

Furthermore, by normalizing, we can assume that the range of $L$ is equal to $[0, 1]$.

**Theorem H.3** (Anthony & Bartlett (1999)). *Let $\mathcal{D}$ be a distribution over graph instances and $\mathcal{A}$ be a class of functions $a : H \to [0, 1]$ with pseudo-dimension $d_{\mathcal{A}}$. Consider $t$ i.i.d. samples $H_1, H_2, \ldots, H_t$ from $\mathcal{D}$. There is a universal constant $c_0$, such that for any $\epsilon > 0$, if $t \geq c_0 \cdot d_{\mathcal{A}}/\epsilon^2$, then we have*

$$\left| \frac{1}{t} \sum_{i=1}^{t} a(H_i) - \mathbb{E}_{H \sim \mathcal{D}} \, a(H) \right| \leq \epsilon$$

*for all $a \in \mathcal{A}$ with probability at least $9/10$.*

The following corollary follows from the triangle inequality.

**Corollary H.4.** *Consider a set of $t$ independent samples $H_1, \ldots, H_t$ from $\mathcal{D}$ and let $\hat{a}$ be a function in $\mathcal{A}$ which minimizes $\frac{1}{t} \sum_{i=1}^{t} a(H_i)$. If the number of samples $t$ is chosen as in Theorem H.3, then*

$$\mathbb{E}_{H \sim D}[\hat{a}(H)] \leq \mathbb{E}_{H \sim D}[a^*(H)] + 2\epsilon$$

*holds with probability at least $9/10$.*

The main challenge is to bound the pseudo-dimension of our given function class $\mathcal{A}$. To do so, we first relate the pseudo-dimension to the VC dimension of a related class of threshold functions. This relationship has been fruitful in obtaining learning bounds in a variety of works such as Lucic et al. (2018); Izzo et al. (2021).

**Lemma H.5** (Pseudo-dimension to VC dimension, Lemma 10 in Lucic et al. (2018)). *For any $a \in \mathcal{A}$, let $B_a$ be the indicator function of the region on or below the graph of $a$, i.e., $B_a(x, y) = sgn(a(x) - y)$. The pseudo-dimension of $\mathcal{A}$ is equivalent to the VC-dimension of the subgraph class $B_{\mathcal{A}} = \{B_a \mid a \in \mathcal{A}\}$.*

Finally, the following theorem relates the VC dimension of a given function class to its computational complexity, i.e., the complexity of computing a function in the class in terms of the number of operations needed.

**Lemma H.6** (Theorem 8.14 in Anthony & Bartlett (1999))**.** *Let $w : \mathbb{R}^\alpha \times \mathbb{R}^\beta \to \{0, 1\}$, determining the class*

$$\mathcal{W} = \{x \to w(\theta, x) : \theta \in \mathbb{R}^\alpha\}.$$

*Suppose that any $w$ can be computed by an algorithm that takes as input the pair $(\theta, x) \in \mathbb{R}^\alpha \times \mathbb{R}^\beta$ and returns $w(\theta, x)$ after no more than $r$ of the following operations:*

- *arithmetic operations $+, -, \times,$ and $/$ on real numbers,*

- *jumps conditioned on $>, \geq, <, \leq, =,$ and $=$ comparisons of real numbers, and*

- *output $0, 1$,*

*then the VC dimension of $\mathcal{W}$ is $O(\alpha^2 r^2 + r^2 \alpha \log \alpha)$.*

Combining the previous results allows us prove Theorem H.1. At a high level, we are instantiating Lemma H.6 with the complexity of *computing* any function in the function class $\mathcal{A}$.

*Proof of Theorem H.1.* First by Theorem H.3 and Corollary H.4, it suffices to bound the pseudo-dimension of the class $\mathcal{A} = L \circ \mathcal{F}$. Then from Lemmas H.5, the pseudo-dimension of $\mathcal{A}$ is the VC dimension of threshold functions defined by $\mathcal{A}$. Finally from Lemma H.6, the VC dimension of the appropriate class of threshold functions is polynomial in the complexity of computing a member of the function class. In other words, Lemma H.6 tells us that the VC dimension of $B_\mathcal{A}$ as defined in Lemma H.5 is polynomial in the number of arithmetic operations needed to compute the threshold function associated to some $a \in \mathcal{A}$. By our definition, this quantity is polynomial in $T(p, k)$. Hence, the pseudo-dimension of $\mathcal{G}$ is also polynomial in $T(p, k)$ and the result follows. $\qquad\square$

Note that we can consider initializing Theorem H.1 with specific predictions. If the family of oracles we are interested in is efficient to compute (which is the case of the predictors we employ in our experiments), then Theorem H.1 assures us that only polynomially many samples are required (in terms of the computational complexity of our function class), to be able to learn a nearly optimal oracle. Furthermore, computing the empirical risk minimizer needed in Theorem H.1 is also efficient for a wide verity of function classes. For example in practice, we can simply use gradient descent or stochastic gradient descent for a range of predictor models, such as regression or even general neural networks.

We remark that our learnability result is in similar in spirit to one given in the recent learning-augmented paper Dinitz et al. (2021). There, they derive sample complexity learning bounds for the different algorithmic problem of computing matchings in a graph (not in a stream). Since they specialize their analysis to a specific function class and loss function, their bounds are tighter compared to the possibly loose polynomial bounds we have stated. However, our analysis above is more general as it allows for a variety of predictors and loss functions to measure the quality of the predictions.

