# OpenReview forum: "Triangle and Four Cycle Counting with Predictions in Graph Streams"
_ICLR.cc/2022/Conference — ICLR 2022 Poster_

### Official Review · Reviewer_FczQ · 2021-11-02

**Correctness:** 4
**Technical Novelty And Significance:** 4
**Empirical Novelty And Significance:** 2
**Recommendation:** 6
**Confidence:** 4

**Main Review:**

Overall the paper is well written and has some nice technical theoretical contributions.
To specifically mention about the strengths I feel those are mostly in terms of the theory and the models.
1) Firstly its about augmenting the ML model with the other streaming models.
2) Theoretically showing that these models have better space complexity as compared to the state of the art.
3) Empirical results for different streaming models along with augmentation of the ML models/oracles.


Here are the points (most of them from the experimental section) that I feel weighs the paper down (can be considered as weakness).
1) The graphs used in the experiments are small in terms of the number of edges as compared to the ones the subgraph counting papers for eg. Bera & Shesadri 2020.
2) Is the space here (in Fig. 1) in terms of the number of edges stored? I.e. is it so that the x-axis denote the number of edges stored?
(It seems so, because in the paragraph above the authors mention that the "only parameter given is a space bound for the number of edges that can be stored")
3) Except for the Powerlaw and Wikibooks graphs (Figures 1f, and 1d), in/for almost all other plots/graphs here, the first coordinate on the x-axis is around 25% of the total edges. Which seems high If I try to compare these values with the ones in the Bera and Sheshadri 2020.
4) Was there any specific reason to select/choose graphs #4 and #30 from Oregon and CAIDA respectively in Fig 1?
I see the results in Fig. 2 for these snapshot graphs with 10% of the edges. However, the amount/fraction of edges used to estimate the number of triangles in other papers for eg. Bera & Sheshadri 2020 and the baselines there, 10% seems high.
5) The algorithm used by the authors, uses a good amount of exact information encoded in the form of the predictors for the experiments. However, it would be good to check if there is any simpler modification of the baseline algorithms of making them stronger by providing these additional information that the authors algorithm use. Otherwise, I am not completely sure if this is a head to head comparison of the algorithms. Or maybe is there a way to say that there is no easier way to incorporate these additional information in the  baseline algorithms?



Additional points:
1) Introduction 3rd para first line -- "patterns do not change quickly over time" --
I think that this depends on the application that you are trying to target. Think of a situation where you keep a track on the number of triangles. You observe tweets on popular trending topics. Add an edge between two entities if they occur together in a tweet. And in this case, I feel the number of triangles can change quickly over time.

2) Table 1: Bera & Sheshadri -- multipass algo:
Can this not be done in a single-pass using -- https://arxiv.org/pdf/1811.08205.pdf -- obviously the complexity changes.

3) Page 3 last para, 4th last line:
The results given by the authors are for general graphs --  would it not be interesting to look to see the relation of space complexity with factors such as arboricity, as observed in the cited papers?

4) It is not clear to me how efficient is it to maintain the oracle/ML model mentioned/used by the authors.
I understand that that it is not the underlying question that the authors are trying to answer.

**Summary Of The Paper:**

The paper proposes a one pass streaming algorithms for estimating the number of triangles in adjacency list and arbitrary order models and 4-cycle in arbitrary edge arrival order. The authors propose algorithms for these streaming models under the assumption of a "heavy" edge oracle/ML model. The paper support theoretical claims with empirical experiments.

**Summary Of The Review:**

In terms of the theoretical contributions, the heavy edge oracle and the information used from that seems to be a good amount of information in the model. Would be worthwhile to check if there is simpler version of the oracle.

In terms of the empirical contributions, the graphs used in the experiments look small in size and space used in terms of the number of edges is more as compared to the other (random edge and random walk based) models. Thus, it would be great to compare the results with the random walk and u.a.r. edge based models with all the algorithms using similar number of edges.

---

### Official Review · Reviewer_tiRU · 2021-11-04

**Correctness:** 4
**Technical Novelty And Significance:** 3
**Empirical Novelty And Significance:** 3
**Recommendation:** 8
**Confidence:** 4

**Main Review:**

The paper is well written, the code is provided, and has clean contributions to the problem of triangle counting, and counting cycles of length 4 in the stream. The mathematical formalizations of the oracles are natural, and lead to a nice framework for analyzing subgraph counting methods using an oracle. The proofs are clean, and rely of course on the formalization and the probabilistic method. My main comment is related to the practical relevance of the proposed scheme that is not clearly illustrated in the experiments. Specifically, the algorithms shown in Table 1 work extremely well in practice. It would have been nice to see the arboricity-dependent method of Bera and Sheshadhri, colorful triangle counting of Pagh and Tsourakakis, and the method of Kallaugher and  Price as competitors.  The proposed method improves the performance assuming an oracle, but the oracle has a computational overhead.  This overhead should be discussed in greater extent for the various learners the authors use in their experiments, and provide some comparison with those well-established methods.   Also a more detailed discussion on how the framework applies to dynamic graphs with both edge deletions and insertions (i.e., turnstile) would be great to have in the main text.   Finally, using bigger datasets would add to the experimental section significantly.

Typo
- In citations "Sofya Vorotnikovs-->Vorotnikova"

**Summary Of The Paper:**

Counting small length cycles in graph streams is an important graph mining primitive. For example, triangles, i.e., cycles of length 3, play an important role in analyzing social networks. Due to the importance of triangle counting, a wide variety of streaming algorithms in different graph steaming models have been proposed over the years.  This work asks the following question: if there is an oracle than can provide information to whether an edge participates in many or few triangles, can we improve the space required?  This work comes as a natural contribution to  a recent series of works on streaming algorithms that assume the existence of an oracle of some form.  The authors formalize the notion of an oracle in two ways, similarly to classification and regression. The former is defined in definition 1.1 where the oracle decides whether or not an edge is "heavy", whereas the latter is termed as the "noisy value oracle" (definition 1.2). The authors contribute to the problem of counting C3s in the incidence and adjacency models assuming a learner, and for C4s in the edge stream model. The authors present first the key idea of their algorithm assuming a perfect oracle, and then show that under certain conditions it works for noisy oracles. For C4s  An overview of their results is given in Table 1, together with a nice overview of the state-of-the-art algorithms.  In terms of experiments, the authors illustrate a variety of learners, learned from in- and out-of-sample.  They show that their methods achieve for some learners non-trivial improvement over various other competitors.

**Summary Of The Review:**

Overall this a well-written paper, with clear contributions to an important problem.

---

### Official Review · Reviewer_oLhV · 2021-11-09

**Correctness:** 3
**Technical Novelty And Significance:** 3
**Empirical Novelty And Significance:** 2
**Recommendation:** 6
**Confidence:** 3

**Main Review:**

Major Strengths of the paper:
1. Theoretical framework with the abstraction of oracles. This might be relevant in the future analysis of such problems
2. Lower bounds for counting problems will be of interest to the community
3. The results do not assume any properties of the graph, unlike some previous works.

Major Issues :
1. Space complexity of oracles is not discussed.

2. Time complexity of the algorithms is not discussed.

3. Is it fair to compare algorithms in terms of the number of passes? Especially when "learning oracle" time is completely excluded in table 1?
For example, the authors say that the McGregor 2016 2-pass algorithm is a combination of  1) first-pass for heavy edge estimation and 2) second pass for triangle counting. Thus it is a particular case of the proposed framework. This exactly highlights the issue in table 1 where we do not involve oracle building time and space complexity which shows the proposed method in a better light than what it actually is.

Other questions and comments :

1. How do predictors used in experiments fit in abstractions used in theorems. i.e., to ask can we compute the $\alpha,\beta$ for valued oracle or $\rho$ for heavy oracles?

2. In the case of an arbitrary stream model, the space complexities are worse in terms of an additive term. However, the authors stress that the one-pass algorithm is the real contribution in these cases. In this particular case, it is even more important to establish that learning the oracle is a cheaper process.

3. Theorems 1.4 and 1.5 do not show any dependence on K. Shouldn't it depend on the noisy-ness of Oracle? For example, what happens when the $K \rightarrow \infinity$? It would mean that Oracle is not at all informative. In such a case with non-informative oracle, it is not clear how do you still get a better space-bound

4. How are the predictor types decided for a particular data set? In "1st graph predictor", does it require O(m) intermediate storage for obtaining exact top 10% edges?

5. GNN with k layers requires access to the k-hop neighborhood of the node under consideration. To apply GNN for an edge, wouldn't you need to store the entire graph ( or subgraph of the same asymptotic size?) to perform the prediction? Also, it appears that for a given edge (u,v), the GNN prediction is applied to edges (u,z) and (v,z) for each z \neq u or v. Wouldn't this be computationally prohibitive?




**Summary Of The Paper:**

The line of research of improving sketching data structures/sampling with the help of learned models is becoming quite popular. This paper follows this line of research and applies this paradigm to the problem of cycle counting in graph streams ( specifically triangles and four-cycles).  It provides a theoretical framework for analyzing Oracle-based problems, analyzing the paradigm's space bounds in cycle counting, and providing lower bounds for the problems.

**Summary Of The Review:**

While some aspects of the paper are nice, such as theoretical framework and lower bounds, etc., the framework's value addition is not clear due to multiple reasons.

1. Authors do not discuss the space complexity of oracles and time ( or number of passes) to train oracle.  This essentially makes me question the comparison in table 1 as indicative of valuable contribution.
2. The experimental results look promising. But again, due to the lack of inclusion of memory required for oracle, it is not clear if the new method actually beats the old techniques in terms of space-complexity.
Also, the predictors that are used might require more auxiliary memory (see points 4,5 above).

Overall, I believe the paper fails to justify the value of the new methods, and including a discussion on the lines of points raised in the review above should help strengthen the paper.  I would be happy to revise my score in case the authors address my concerns.

Update: I have updated my score after the revision provided by the authors. There are some good theoretical contributions in this paper which I believe will be useful. The idea of using machine learning oracles with sketching algorithms is, in general, a established idea which deserves exploration in different problem. For this particular problem of counting, I am still a bit reserved on applicability of this framework to the specific problem at hand when accounting for all the meta costs (such as oracle learning etc). However, I am still of the opinion that this paper is worth the acceptance for its overall value to the community.

---

### Author Response · Authors · 2021-11-15
**Thank you to all reviewers**

We thank the reviewers for their valuable feedback. Answers are given in a response to each review. We have uploaded a revised version of the paper, with newly added segments marked in blue for the convenience of the reviewers.

---

### Decision · Program_Chairs · 2022-01-20

**Decision:**

Accept (Poster)

**Comment:**

The paper proposed a novel one-pass efficient streaming algorithm for estimating the number of triangles and four cycles. The concerns raised by reviewers were nicely addressed in the rebuttal and all the reviewers agree that the paper is above bar for publication.